# NanoPlex: a universal strategy for fluorescence microscopy multiplexing using nanobodies with erasable signals

Nikolaos Mougios[1,2], Elena R. Cotroneo[3], Nils Imse[3], Jonas Setzke [2], Silvio O. Rizzoli [1,4], Nadja A. Simeth [3,4], Roman Tsukanov [5] & Felipe Opazo [1,2,6] ✉

Fluorescence microscopy has long been a transformative technique in biological sciences. Nevertheless, most implementations are limited to a few targets, which have been revealed using primary antibodies and fluorescently conjugated secondary antibodies. Super-resolution techniques such as Exchange-PAINT and, more recently, SUM-PAINT have increased multiplexing capabilities, but they require specialized equipment, software, and knowledge. To enable multiplexing for any imaging technique in any laboratory, we developed NanoPlex, a streamlined method based on conventional antibodies revealed by engineered secondary nanobodies that allow the selective removal of fluorescence signals. We develop three complementary signal removal strategies: OptoPlex (light-induced), EnzyPlex (enzymatic), and ChemiPlex (chemical). We showcase NanoPlex reaching 21 targets for 3D confocal analyses and 5–8 targets for dSTORM and STED super-resolution imaging. NanoPlex has the potential to revolutionize multi-target fluorescent imaging methods, potentially redefining the multiplexing capabilities of antibody-based assays.

Fluorescence microscopy has become a beneficial application for cell biologists, partly because specific targets can be revealed on a complex biological sample. At present, fluorescence imaging systems can routinely discriminate four different colors/channels (i.e., emission on the "blue", "green", "red", and "deep-red" spectra) some setups can detect near-infrared light (>700 nm), adding a fifth channel and others are assisted by spectra unmixing software, making discrimination of 6 to 7 targets possible[1]. The most common strategy to achieve multi-target labeling in biological samples is by indirect immuno-fluorescence (IF), which relies on primary antibodies (1.Abs) that recognize a POI selectively, followed by fluorescently-labeled secondary antibodies (2.Abs) binding to unique epitopes present on the species or iso-type of the 1.Ab (e.g., binding to mouse IgGs or chicken IgYs, etc.). Therefore, to reveal, for example, three targets using classical IF, it is necessary to have three specific 1.Abs, each from a different species, to allow the specific detection by species-specific fluorescent 2.Abs.

Although this strategy has resulted in important discoveries, detecting a higher number of targets (multiplex) is often desired (e.g., in the growing field of single-cell proteomics[2]). An initial attempt to localize six targets in the same sample was performed over two decades ago, where antibodies were stripped from the sample using chemical denaturation[3] or photobleached[4] before performing a new IF cycle. These cycles allowed the image of more than 10 targets to create

[1]Institute of Neuro- and Sensory Physiology, University Medical Center Göttingen, Göttingen, Germany. [2]Center for Biostructural Imaging of Neurodegeneration (BIN), University of Göttingen Medical Center, Göttingen, Germany. [3]Institute for Organic and Biomolecular Chemistry, University of Göttingen, Göttingen, Germany. [4]Cluster of Excellence "Multiscale Bioimaging: from Molecular Machines to Networks of Excitable Cells" (MBExC), University of Göttingen, Göttingen, Germany. [5]III. Institute of Physics - Biophysics, Georg August University, Göttingen, Germany. [6]NanoTag Biotechnologies GmbH, Göttingen, Germany. ✉e-mail: fopazo@gwdg.de

the so-called toponome maps[5]. Several variations of these concepts were further explored. For instance, applying strong detergents[6,7], fluorophore-inactivating chemicals[8–10], chaotropic salts[11,12], extreme pH[6,13] or microwave heating of the sample[14,15] have been used to eliminate the fluorescence signal, enabling subsequent staining and imaging cycles. However, most of these techniques have the potential to compromise cellular structures. Alternative techniques using single-stranded DNA (ssDNA) for barcoding signals, like immuno-SABER[16] or CODEX[17], focus on amplifying signals on multi-target, resulting in large-area histological imaging (for review, see ref. [18]). The resolution limit, and for some techniques, the deterioration of the sub-cellular ultrastructure and epitopes induced by harsh treatments, compromises the precision of molecular localization and abundance determination for every imaging cycle. Therefore, these multiplexing strategies may not be feasible when studies require evaluations of protein levels or demand high fidelity in subcellular and molecular localization.

With the increasing adoption of super-resolution microscopy techniques[19,20], several strategies to obtain images from multiple targets have also been developed. Currently, the most multi-target capable super-resolution microscopy techniques are based on Exchange-PAINT[21], where antibodies (or other affinity-based tools) are coupled to a ssDNA. The sequence of this ssDNA provides a unique identifier to this antibody and its target, which can be later revealed by a complementary ssDNA conjugated to a fluorophore. This strategy allows labeling multiple antibodies and targets decorated with different ssDNA sequences and calculating the emitter position in $x$, $y$, and $z$ dimensions with a few nanometers of precision. This strategy has achieved 9-plex super-resolved images in a cellular context[22], and only recently, more complex variations termed SUM-PAINT and FLASH-PAINT that used multi-barcoding ssDNA has demonstrated 30-plex and 13-plex super-resolution images respectively[23,24]. While Exchange-PAINT and now SUM-PAINT allow multiplexed imaging with a high degree of detail in cells, the technology is not straightforward to implement in PAINT-naïve laboratories since dedicated hardware, software, and specialized expertise are required for optimal results.

Therefore, we have established a simpler and more universally applicable approach to image multiple targets using any conventional light microscope. We also demonstrate our technique with two widely-used super-resolution techniques: direct STochastic Optical Reconstruction Microscopy (dSTORM) or Stimulated Emission Depletion (STED) super-resolution microscopy. The methodology exploits the extensive diversity of existing antibodies and utilizes engineered secondary nanobodies (2.Nbs). These 2.Nbs allow not only a one-step multitarget and species-independent immunofluorescence (IF)[25], but, as presented here, iterative imaging by selectively removing the fluorescence reporter on the 2.Nbs (NanoPlex). Our signal-erasing strategy differs from other iterative methods by refraining from using detergents, high-energy bleaching of fluorescence, microwave heating, harsh chemicals, or extreme pH. Instead, our approach involves the modification of 2.Nbs with several mild cleavage options, ensuring the specific and efficient removal of soluble fluorescent moieties. We initially designed a wavelength-specific photolabile molecule allowing fast and region-specific light-induced cleavage (OptoPlex), which could be applied to both fixed and live samples. A second approach minimizes the potential UV-light sample damage by using a specific enzymatic cleavage (EnzyPlex), which can be applied systematically to larger areas in living or fixed specimens. Finally, to have a very efficient, cheap, and universally usable strategy, we developed 2.Nbs whose fluorescent group can be removed using a commercially available chemical (ChemiPlex). Implementing these strategies will allow universal multiplex capability to most antibody-based applications (e.g., Western blots), thereby simplifying and expanding the detection of multiple targets across various techniques and fields.

## Results

### Creating nanobodies with UV-erasable fluorescence signal (OptoPlex)

We aimed to engineer fluorescent nanobodies to facilitate the gentle and rapid removal of their fluorophores post-imaging, enabling new staining and imaging cycles to eventually reveal multiple targets through several iterations (Fig. 1a). The strategy exploits 2.Nbs' simplicity for directed modifications and their advantageous one-step IF, resulting in species-independent multi-staining capabilities[25].

Our initial strategy was to achieve fast and selective removal of fluorophores from 2.Nbs and cleave them with light, an approach we call OptoPlex. For this, we first created and characterized a light-responsive tag (LRT) molecule carrying both the ALFA-tag peptide[26] and a maleimide handle for labeling (Fig. 1b) (for synthesis details, see Methods and Supplementary Fig. 1). The distances between the functional units (linkers) were kept as short as possible to limit resolution loss while simultaneously ensuring sufficient space for binding of the anti-ALFA-tag nanobody (NbALFA). As a photolabile unit, we used an *ortho*-nitrobenzene (ONB), which has been previously used in cells[27]. ONB has its lowest electronic transition at wavelengths ($\lambda$) around 365 nm, showing no absorption above 400 nm (Fig. 1c), which allows it to be used $\lambda$-orthogonally to the excitation lights commonly employed in biological imaging. When LRT is illuminated with 365 nm, we can follow its cleavage due to a change in absorption (Fig. 1d, Supplementary Figs. 2, 3). Then, LRT was site-specific conjugated to the C-terminus of 2.Nbs, bringing a photo-cleavable linker between the 2.Nb and the ALFA-tag (i.e., the Opto-2.Nbs).

The first cellular test for the Opto-2.Nbs was to erase the signal in a defined region of the sample. For this, we stained the intermediate filament vimentin of U2OS-Nup96-GFP cells[28], using one-step IF by pre-forming the complex between 1.Ab (anti-vimentin), Opto-2.Nb and NbALFA-Atto643 in a 1:3:4 molar ratio (See Methods and Supplementary Table 3). After mounting the sample in a flow chamber (Supplementary Fig. 4), widefield images were acquired, and only a small region was then illuminated with 365 nm LED light to observe the loss of the immunostained vimentin signal in the illuminated region (Fig. 1e). Importantly, the signal from enhanced green fluorescent protein (EGFP) on the nuclear pores (Nup96) was unaffected by the UV illumination (Fig. 1e, Supplementary Fig. 5a).

For a more quantitative assessment of cleavage, we stained vimentin again on U2OS-Nup96-GFP using one-step IF with Opto-2.Nbs, and imaged under laser scanning confocal microscopy. After the first confocal image, the sample was continuously illuminated using ~60 mW/cm² of 365 nm LED light. New images were acquired at different time points for 15 min (Fig. 1f). Maximal signal removal was achieved after ~5 min of illumination (Fig. 1g). EGFP signal was not affected even after 15 min of continuous illumination with 365 nm light. In addition, we stained vimentin with 2.Nbs directly labeled with Atto643 (not cleavable) and observed no significant loss of vimentin signal when illuminated using ~60 mW/cm² of 365 nm LED light, confirming that the specific signal erasing is achieved by photo-cleavage (Supplementary Fig. 5b).

After these promising results, we performed an iterative light-induced removal of fluorophores (OptoPlex) and successfully obtained images of multiple targets on the same sample (Fig. 1h). For this, we pre-mixed 1.Abs with Opto-2.Nbs and each pre-mixture carried a different fluorophore (AZDye568 and Atto643, details on Supplementary Table 3). One-step IF allowed us to simultaneously reveal the tubulin filaments and clathrin (Fig. 1h, cycle 1). After acquiring a confocal image for each target, the field of view was illuminated continuously for 15 min with 365 nm light while perfusing with Washing-Buffer (Supplementary Table 2) to remove the cleaved fluorescent groups. Subsequently, one-step IF of vimentin and peroxisomes (cycle 2), imaging, and photocleaving followed by a third cycle to obtain images of nuclear speckles and nuclear pores (cycle 3) (Fig. 1h).

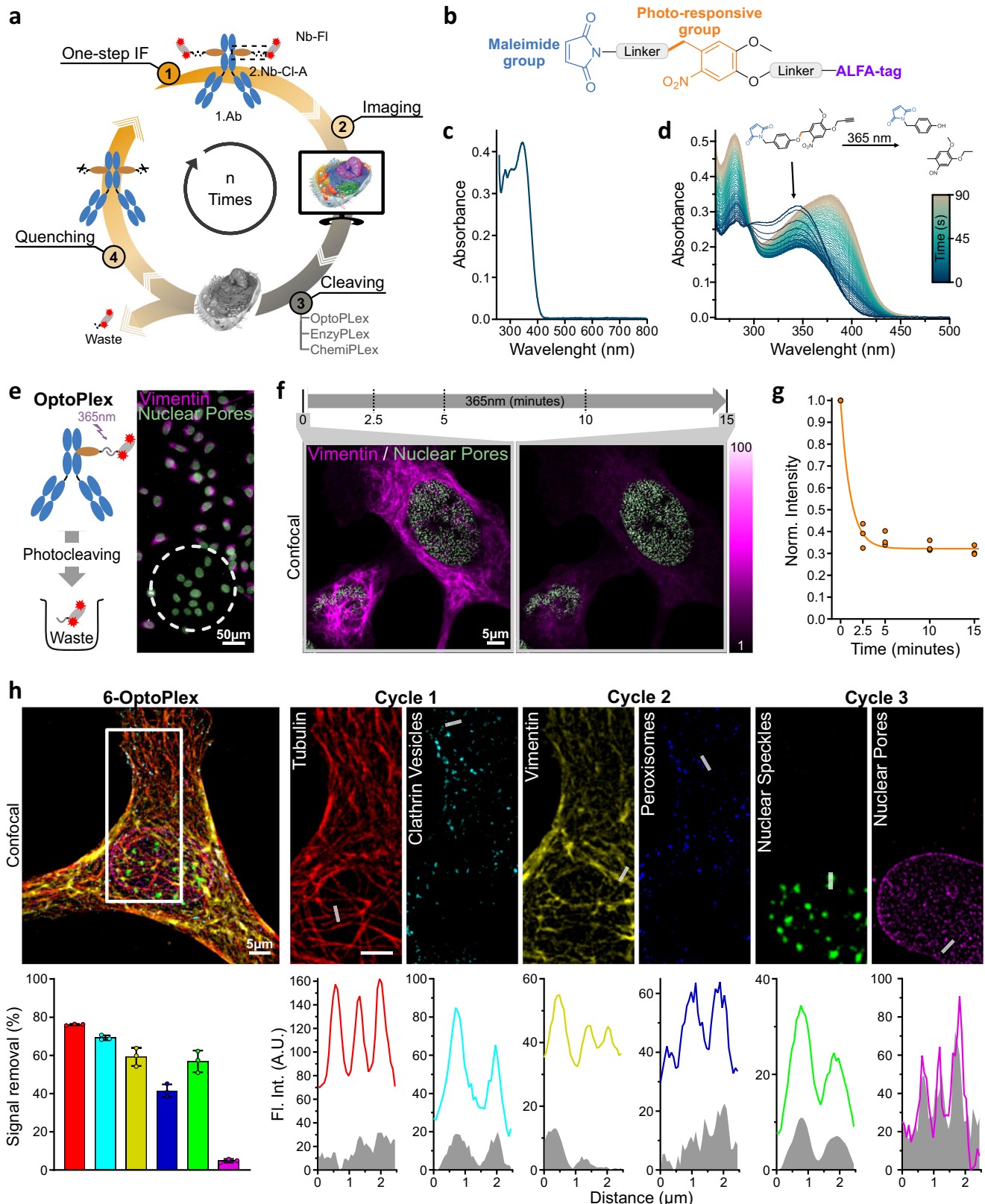

Confocal Images were acquired after every photo-cleavage, revealing that signal removal was target specific. The poorest removal of signal seems to correlate to membrane-associated proteins like PMP70 on peroxisomes (Fig. 1h and Supplementary Fig. 5c). Nevertheless, the remaining signal was not high in absolute terms, and by mild thresholding, it was simple to obtain a 6-plex image under confocal microscopy, showing the potential, the region-specific signal removal, and flexibility of OptoPlex as a multiplexing strategy.

**Using a specific protease to erase fluorescence signal (EnzyPlex)**
To avoid the potential photodamage of biological specimens that UV irradiation might cause in the OptoPlex workflow, we designed a

**Fig. 1 | A universal methodology for multiplexed light microscopy based on UV-light cleavable 2.Nb. a** Overview of NanoPlex. Step 1, one-step IF, takes place using pre-formed complexes of a 1.Ab, a 2.Nb functionalized with a cleavable (Cl) linker followed by an ALFA-tag (2.Nb-Cl-A) and a fluorescently labeled NbALFA. The protein of interest is imaged in Step 2, while in Step 3, three different approaches can be used to cleave the linker and erase the fluorescent signal. Potential remaining reactive components are quenched in Step 4 before re-initiating Step 1. **b** Light Responsive Tag (LRT) comprises a maleimide group followed by a photo-responsive group and an ALFA-tag. **c** LRT absorbance for wavelengths between 260 and 800 nm (50 μM, DMSO). **d** Tracking the absorption of TLR for wavelengths between 260 and 500 nm during photo-induced cleavage with 365 nm light. The transition of the absorbance was recorded approximately every second while illuminating with 365 nm light for ~93 s. **e** Light-induced cleaving and removal of fluorescent groups (OptoPlex) from pre-formed complexes of 1.Ab and 2.Nb conjugated to LRT (Opto-2.Nb) and complexed with NbALFA-Atto643. U2OS-Nup96-GFP cells (green) were immunostained against vimentin (magenta), and region-specific cleaving (white dotted circle) was achieved after constant illumination with 365 nm light for 15 min. **f** Confocal images from U2OS-Nup96-GFP cell (green) were labeled against vimentin (magenta) using OptoPlex complexes. The samples were illuminated with 365 nm light for 15 min, and images were recorded at 0, 2.5, 5, 10, and 15 min after applying 365 nm light. **g** The fluorescent intensity recorded at each time point was normalized to the values before applying 365 nm light (3 independent experiments, the line represents a one-phase decay fit of the data points). **h** Confocal 6-Plex image from U2OS-Nup96-GFP after 3 iterative cycles of OptoPlex and the respective signal removal efficiency for each target. The NbALFA-Atto643 was used for tubulin and vimentin, while the NbALFA-AZDye568 was used for clathrin, peroxisomes, and nuclear speckles. EGFP signal was acquired from Nup96-GFP (NPC). Plot profiles depict the fluorescence intensity across the denoted line for each target before (color) and after (grey) signal removal. Staining details are in Methods and Supplementary Table 3. Bar graphs indicate the mean ± SD; n = 3 cells.

second approach, in which we produced 2.Nbs fused to a small ubiquitin-related modifier (SUMO) substrate derived from the plant *Brachypodium distachyon* (bdSUMO) followed by an ALFA-tag. bdSUMO is specifically and efficiently cleaved by the protease bdSENP1, obtained from the same organism[29] (Fig. 2a). We first confirmed the strategy by incubating the enzymatically cleavable 2.Nbs (Enzy-2.Nbs) with bdSENP1 and loaded them in a polyacrylamide gel to verify the proteolytic product (Fig. 2b, Supplementary Fig. 6a). To benchmark EnzyPlex in cells, we then stained vimentin using 1.Ab and Enzy-2.Nbs on U2OS-Nup96-GFP cells and found that cleaving the fluorescent group using bdSENP1 resulted in erasing ~87 ± 2% (mean ± SD) of the signal within ~15 min (Fig. 2c, d). Control experiments revealed that samples stained with non-cleavable fluorophores on the 2.Nbs resulted only in ~8 ± 7%, (mean ± SD) signal reduction (Supplementary Fig. 6b). To challenge the ability of EnzyPlex to reveal multiple targets, we performed the first conventional confocal microscopy, where we labeled two targets simultaneously, each labeled with a different color (Atto643, and AZDye568, details in Supplementary Table 4), followed by two full experimental cycles including bdSENP1 cleaving, one-step staining and imaging of two new targets per cycle. This strategy delivered a simple 6-plex (2-colors × 3-cycles) confocal imaging (Fig. 2e), resulting in a marginal remaining signal after enzymatic cleavage (Supplementary Fig. 6c).

Finally, we challenged the EnzyPlex strategy to perform multi-color dSTORM imaging using a custom-made optical setup (Supplementary Fig. 7). Single-channel dSTORM was employed for 5 iterative cycles of EnzyPlex, for which we obtained super-resolved images of microtubules, clathrin, vimentin filaments, peroxisomes, and nuclear pore complexes (Fig. 2f) using the same fluorophore (JF635b[30]) resulting in multiplexed dSTORM images of 5 targets with high average localization precision of 12.9 ± 2.5 nm (mean ± SD), estimated using the Nearest Neighbor (NeNa) algorithm[31]. Performing the EnzyPlex dSTORM experiments took per target ~1 h of staining, ~0.5 h of imaging, and ~0.3 h of cleaving and washing, summing up to ~2 h per target. However, there is no practical limit to stopping in 5 cycles or performing more colors per cycle[32,33], suggesting that EnzyPlex could achieve a much higher level of multiplexability in complex applications like dSTORM.

Signal removal using EnzyPlex was superior to the removal obtained by OptoPlex. However, cleavage efficiency remained not homogenous for all targets (Supplementary Fig. 6c). This inhomogeneity in cleaving could be the result of bdSENP1 failing to find its substrate due to steric hindrance or simply a limited diffusion within biological samples after the imaging cycles.

## Using redox chemistry to remove fluorescence (ChemiPlex)
This strategy brings a disulfide bond between the 2.Nb and the ALFA-tag, providing a fast, global, and even simpler way to erase signals. For this, we conjugated the N-terminus to the ALFA-tag peptide with succinimidyl 3-(2-pyridyldithio)propionate (SPDP-ALFA). We used this SPDP-ALFA and coupled it to the reactive thiol on a cysteine on the C-termini of 2.Nbs, resulting in a disulfide bridge between the nano-body and the ALFA-tag (Chemi-2.Nbs). The release of the ALFA-tag with its associated fluorescent group is achieved by simply applying a small reducing agent (Fig. 3a). Similarly to previous strategies, we performed one-step IF by pre-mixing the 1.Ab anti-vimentin, with Chemi-2.Nbs and NbALFA-Atto643. Confocal microscopy images were acquired before and after adding 10 mM tris(2-carboxyethyl)phosphine (TCEP) as a reducing agent (Fig. 3b, c). After exposing the sample to TCEP-Buffer (Supplementary Table 2) for 15 min, ~95% of the signal was removed (Fig. 3c). When the Enzy-2.Nb and NbALFA-Atto643 were used in control experiments in the presence of TCEP, only bleaching could be observed when imaged for 15 min (Supplementary Fig. 8a). This demonstrates that the TCEP-Buffer does not affect either the affinity-based probes or the fluorophore (Supplementary Fig. 8b). We continued by testing 6-targets in confocal imaging, as performed for OptoPlex and EnzyPlex, but this time successively erasing the signal 5 times, using a single channel per ChemiPlex round (Fig. 3d). This attempt resulted in a very efficient and homogenous cleavage throughout the six cycles (Fig. 3d and Supplementary Fig. 8b).

ChemiPlex resulted in a simple and effective strategy to remove the signal for all targets tested; thus, we decided to try ChemiPlex cycles under super-resolution microscopy. However, due to the need for reducing agents for signal erasure, this approach won't work under conventional dSTORM conditions, where reducing agents are needed in the buffer to help with the blinking of the fluorophores. Therefore, we turned to test ChemiPlex under the imaging challenges of STED super-resolution microscopy. To achieve several cycles of high-energy illumination and iterative stainings, we adapted our Imaging-Buffer (Supplementary Table 2), where we included PCA/PCD and Trolox scavengers during confocal and STED imaging acquisition to minimize detrimental effects caused by reactive oxygen species (ROS) generated during high energy excitation of fluorophores[34,35]. Additionally, we bathed the sample after each TCEP cleavage step with N-Ethylmalimide (Thiol-Quenching, Supplementary Table 2) to block potentially generated reactive thiols before the next staining cycle. Finally, after the complex formation between 1.Ab and fluorescent Chemi-2.Nbs, we added an excess of unconjugated (dark) 2.Nbs to block any potential epitope available on 1.Ab from previous cycles (see Methods). With all these precautions in place (see scheme in Supplementary Fig. 9), we stained (Supplementary Table 7), imaged, and finally erased the signal to obtain eight targets on the same sample, using the best channel in our STED setup (Fig. 4a). After each TCEP cleavage, only a marginal background signal remains (Fig. 4c, Supplementary Fig. 9), allowing us to proceed with the next staining and imaging cycle. We looked at different subcellular structures, including various organelles like peroxisomes (PMP70), mitochondria (TOM20), and Golgi (GALNT2), but also abundant filaments (tubulin and vimentin), inside the nucleus

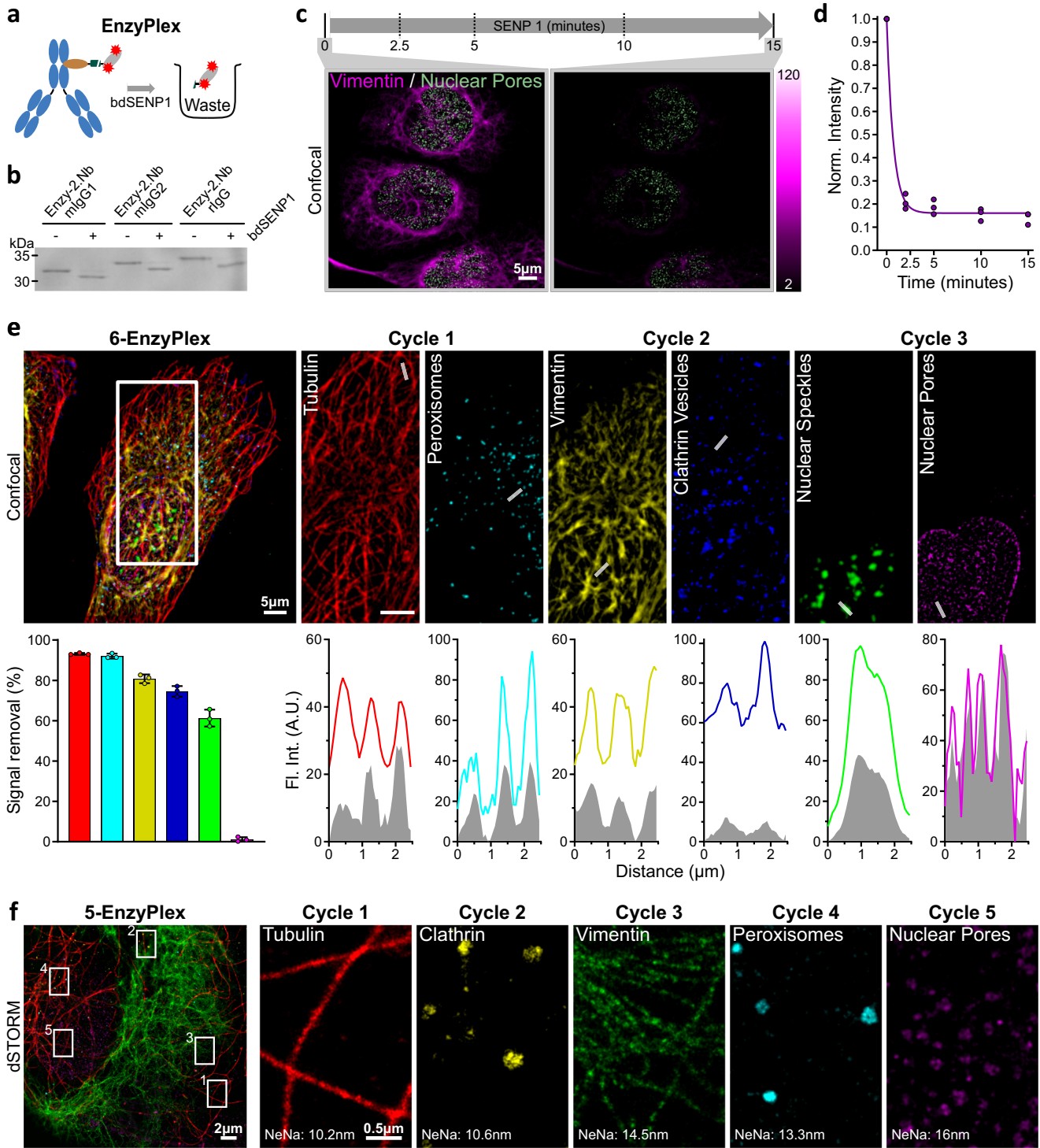

**Fig. 2 | Enzymatic removal of fluorescent signal (EnzyPlex). a** Schematic of EnzyPlex cleaving process. Pre-formed complexes of 1.Ab and fluorescent Enzy-2.Nbs can be cleaved with bdSENP1 protease. **b** SDS-PAGE of different Enzy-2.Nbs before (-) and after (+) the addition of 1 μM bdSENP1. **c** Confocal images from U2OS-Nup96-GFP cells labeled against vimentin (magenta) using complexes of 1.Abs and Enzy-2.Nb carrying the Atto643. The sample was treated with 1 μM of bdSENP1 for 15 min, and vimentin's fluorescent signal was recorded at 0 (before cleavage), 2.5, 5, 10, and 15 min upon bdSENP1 application. **d** The fluorescent intensities at each time-point were normalized to the intensity values before the application of bdSENP1. **e** Confocal 6-Plex image from U2OS-Nup96-GFP cells undergoing 3 iterative cycles of EnzyPlex and 2-channel confocal imaging per cycle. Bar graph displays the mean ± SD of the signal removal efficiency for each target (*n* = 3 cells). The NbALFA-Atto643 revealed tubulin and vimentin, while NbALFA-AZDye568 fluorophore was used for clathrin, peroxisomes, and nuclear speckles. GFP signal was acquired from Nup96-GFP (NPC). Plot profiles depict the fluorescence intensity across the denoted line for each target before and after signal removal. Staining details are in Methods and Supplementary Table 4. **f** 5-EnzyPlex dSTORM image from U2OS cells upon 5 iterative cycles. The Janelia Fluor JF635b was employed for single molecular localization imaging of alpha-tubulin (1), clathrin (2), vimentin (3), peroxisomes (4) and Nup96-EGFP on nuclear pores (5) with average localization precision (nearest neighbors distance estimator-NeNa) of 10.2, 10.6, 14.5, 13.3 and 16 nm, respectively. Details on stainings are in Supplementary Table 5.

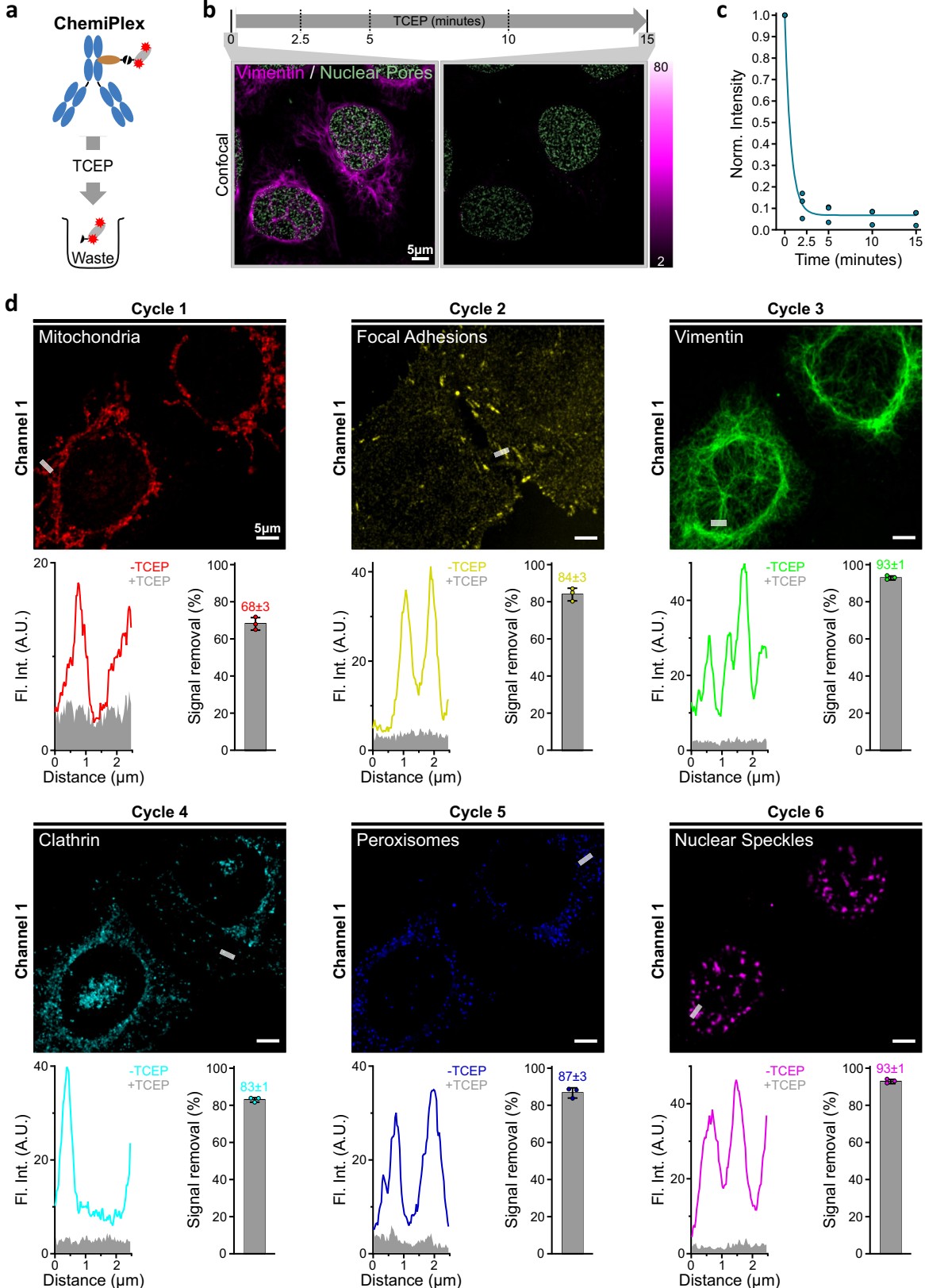

(transcription factor on the nucleoli), cytoplasmic clathrin and nuclear pore complexes at the nuclear envelope. The resolution achieved was as expected from a commercial setup, and it was not compromised during the progress of cycles, allowing us to clearly observe the pores on NPCs stained and imaged in the last cycle (Fig. 4b). As an additional control for sample integrity, we stained the last cycle using a primary

nanobody directly conjugated to Atto643 to reveal the Nup96-GFP (nuclear pores). This nanobody binds to a structural epitope on GFP; thus, the specific and high-resolved image suggests that the GFP structure was not damaged after the seven cycles of exposure to the TCEP-Buffer and to the powerful light imaging during the ChemiPlex-STED procedure. The number of cycles has been limited only by

**Fig. 3 | Erasing the fluorescent signal with a reducing agent (ChemiPlex).**
**a** Schematic of ChemiPlex cleaving process. The fluorescent groups in pre-formed complexes of 1.Ab and Chemi-2.Nb, can be cleaved and removed after treatment with TCEP. **b** Laser scanning confocal images from U2OS-Nup96-GFP cells labeled against vimentin (magenta) using preformed complexes with Chemi-2.Nb carrying the Atto643. The sample was treated with TCEP for 15 min, and the vimentin's fluorescent signal was recorded at 0 (before adding TCEP), 2.5, 5, 10, and 15 min upon TCEP application. **c** The fluorescent intensities recorded at each time point were normalized to the intensity values before applying TCEP. **d** Representative 6-Plex confocal from U2OS-Nup96-GFP cells after application of 6 iterative cycles of ChemiPlex. The plot profiles depict the fluorescence intensity across the denoted line for each target before and after signal removal, while the bar graph shows the signal removal after treatment with TCEP (mean ± SD. $n = 3$ independent regions). Images from mitochondria, focal adhesions, vimentin, clathrin vesicles, peroxisomes, and nuclear speckles were recorded using the same Atto643, fluorophore; for all staining details, see Methods and Supplementary Table 6.

finding strong and specific 1.Abs that can work under the same sample preparation conditions (i.e., same fixation chemistry, duration, and permeabilization used). Therefore, we envision our method going beyond the 8-Plex STED we obtained using a single-color STED strategy.

## Imaging multiple proteins on primary hippocampal neurons

To showcase the simple implementation of multitarget imaging using NanoPlex, we performed ChemiPlex to acquire confocal z-stacks for 7 cycles while imaging 3 channels per cycle, resulting in 21 different targets in 3D on the same neuronal culture (Fig. 5, Supplementary Fig. 10). For this, we perform all pre-mixtures between 1.Ab and 2.Nb as detailed in Supplementary Table 8. We revealed pre- and post-synaptic targets as well as filament and filament-associated proteins, endosomes, peroxisomes, and astrocytes. We kept the scavengers in the Imaging-Buffer (Supplementary Table 2) to minimize ROS damage. Cleaving was performed with TCEP-Buffer (Supplementary Table 2), and the resulting reactive thiols were neutralized using Thiol-Quencher buffer (Supplementary Table 2). The 2.Nb blocker (Methods and Supplementary Table 11) was included with the immunostaining step at every cycle.

We then focused on synaptic targets and studied the individual behaviors of nine synaptic proteins in more detail. We stained primary neurons with markers for excitatory and inhibitory synapses (vGlut1, PSD-95, and vGAT, gephyrin, respectively), as well as other synaptic markers like alpha-Synuclein (aSyn), Synapsin-1 (Syn1), Synaptotagmin-1 (Syt1), VAMP2 and Rab3a (Fig. 6a, Supplementary Table 9). To confirm that our 9-ChemiPlex provided relievable results, we first analyzed the colocalization (Pearson's correlation) of proteins expected to correlate, e.g., between the excitatory presynaptic vGlut1 and the excitatory postsynaptic PSD-95, or the inhibitory presynaptic vGAT with the inhibitory postsynaptic gephyrin. This analysis provided the correlations of 0.69 ± 0.05 and 0.52 ± 0.12 (mean ± SEM), respectively (Fig. 6b). The opposite trend was found when correlating as control inhibitory elements with excitatory markers (Fig. 6b). As an internal control for potential artefactual correlations due to residual signal remaining from previous cycles, we analyzed Gephyrin and PSD-95 signals, which were obtained using the same fluorophore in different cycles. Results suggest that Gephyrin and PSD-95 have a Pearson´s correlation of -0.15, obtained from a residual signal, autofluorescence, and the simple chance of excitatory and inhibitory boutons being close by or within the confocal resolution. Additionally, we controlled if these are the expected correlation values if using conventional 2-color confocal microscopy on the same targets using the same antibodies (Supplementary Fig. 11). This control resulted in pre-and post synaptic proteins like vGlut and PSD-95 showing Pearson´s of 0.65 ± 0.05 and of 0.60 ± 0.04 for gephryrin and vGAT. Similar for a low expected correlation, where gephyrin and PSD-95 resulted in 0.13 ± 0.04 and 0.12 ± 0.04 for vGlut and vGAT. These results corroborated that the correlation values obtained by ChemiPlex do not deviate from the values obtained with conventional 2-color imaging, even if we try different fluorophore combinations (Supplementary Fig. 11).

Through a careful analysis of the correlations among these nine synaptic proteins, we could confirm several observations reported in earlier studies (Fig. 6c and Supplementary Fig. 12). For instance, in our experiments, aSyn signal correlates positively to VAMP2 as demonstrated before[36], but aSyn seems to be absent or in meager amounts on inhibitory synapses defined by gephyrin and vGAT as another work has suggested[37]. Similarly, Synapsin-1, a known protein for keeping synaptic vesicles (SVs) at presynapses, highly correlates to aSyn as recently described in ref. 38. Overall, results and conclusions from multiple labs and years of work could be reproduced in a single NanoPlex experiment.

Recently, several key synaptic functions like neurotransmitter release and receptor signaling have been associated with proteins and organelles engaging in liquid-liquid-phase separation (LLPS)[39]. To analyze this process, we treated primary cultured neurons with 1,6-hexanediol (HEX), which is known to interfere with liquid-liquid phase formation[40], and performed ChemiPlex, to again reveal these nine synaptic targets and detect any changes in their Pearson's correlation coefficients (Fig. 6d, Supplementary Fig. 12). The results suggest that Syn1, which is a main driver of LLPS in synapses, loses correlation to aSyn, which has been described in association with SVs and Syn1[38]. The simplest interpretation is that these two soluble (not transmembrane) molecules separate, as the LLPS-based SVs cluster is disturbed by the HEX treatment. Therefore, it suggests that aSyn correlation with Syn1 is more affected by HEX, than its correlation with SVs. On the contrary, Syn1 increases its correlation with three SV proteins, VAMP2, Syt1, and Rab3a (Fig. 6d, Supplementary Fig. 12), implying that this molecule remains attached to SVs, albeit its ability to drive LLPS is lowered in the presence of HEX (Fig. 6c, d). In general, correlations between SV proteins, including vGlut, Syt1 and VAMP2, are lowered, indicating that the SV clusters are loosened, resulting in lower signals with poorer correlations (Fig. 6d, Supplementary Fig. 12). Surprisingly, vGlut1 loses correlation to other transmembrane SV proteins like VAMP2, and Syt1 (Fig. 6d, Supplementary Fig. 12). While this could be explained by the simple effect of HEX scattering SVs resulting in a drop in correlation, it could also show that vGlut1 is located more strictly at SVs while VAMP2 and Syt1 can also be found on patches at on the plasma membrane of presynapses[41]. These results confirm that LLPS keeps vesicles and related proteins together at presynapses.

In addition, we observed that loosening the LLPS results in increased correlation between the gephyrin inhibitory and PSD-95 (inhibitory and excitatory postsynaptic proteins whose assemblies have been linked to LLPS[40,42]) (Fig. 6d, Supplementary Fig. 12). For a more precise understanding on this unexpected effect, we focused on excitatory (PSD-95-positive) and inhibitory (gephyrin-positive) synaptic sites and measured the fluorescence intensity (at the synapse) of all nine targets before and after HEX treatment. An interesting example is PSD-95, where fluorescence intensity at excitatory sites increases and exhibits a "sharper" profile upon HEX treatment, supporting recent high-resolution imaging findings[43]. Other proteins, like Syt1, remain stable in both excitatory and inhibitory sites upon HEX addition (Fig. 6e). On the contrary, vGlut1 is moderately lost from excitatory sites upon HEX, while our data suggest a stronger depletion from inhibitory sites (Fig. 6e), implying that vGlut1 (excitatory marker) might be initially localized in inhibitory synapses in hippocampal cultures, as recently proposed[23]. Overall, our synaptic analysis indicates that inhibitory synaptic sites are more affected by HEX compared to excitatory synaptic sites (Supplementary Fig. 12)

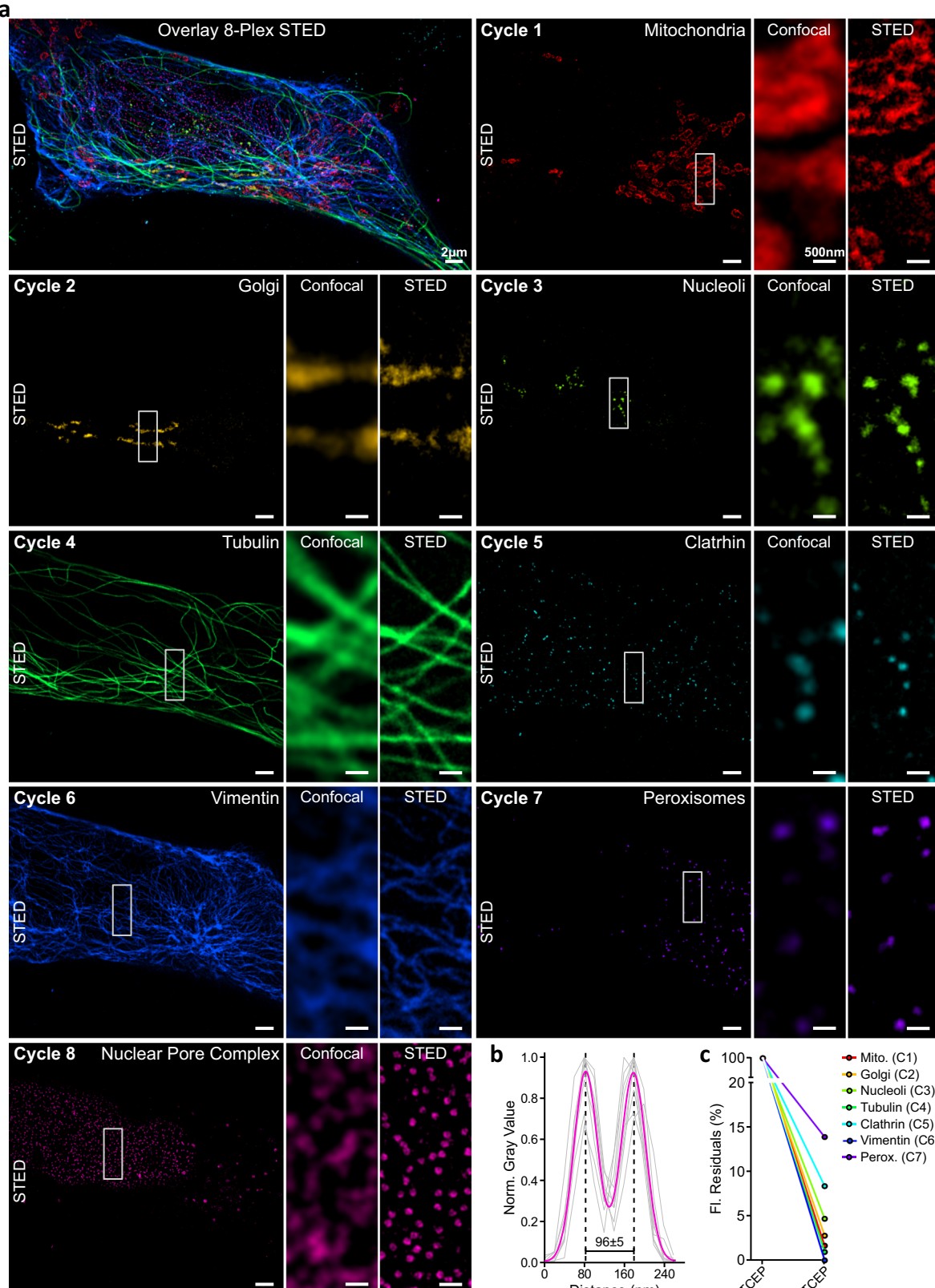

**Fig. 4 | Iterative ChemiPlex cycles for 8-Plex STED imaging using a single fluorophore. a** Overlay of STED images acquired after 8 repetitive cycles of ChemiPlex. From cycle 1 to cycle 8, Chemi-2.Nb carrying Atto643 were employed to reveal mitochondria (Cycle 1; C1), Golgi (C2), nucleoli (C3), tubulin (C4), clathrin (C5), vimentin (C6), peroxisomes (C7) and nuclear pore complexes (C8) in both confocal and STED resolutions. **b** Plot of normalized fluorescence intensities profile of 9 nuclear pores, in gray. The fit of the average intensity profile is plotted in magenta and the diameter was calculated as $96 \pm 5$ nm (mean $\pm$ SD). **c** Fluorescent intensity residuals in confocal imaging for each cycle of ChemiPlex (C1 to C7, full images on Supplementary Fig. 9) were normalized to signal before cleaving.

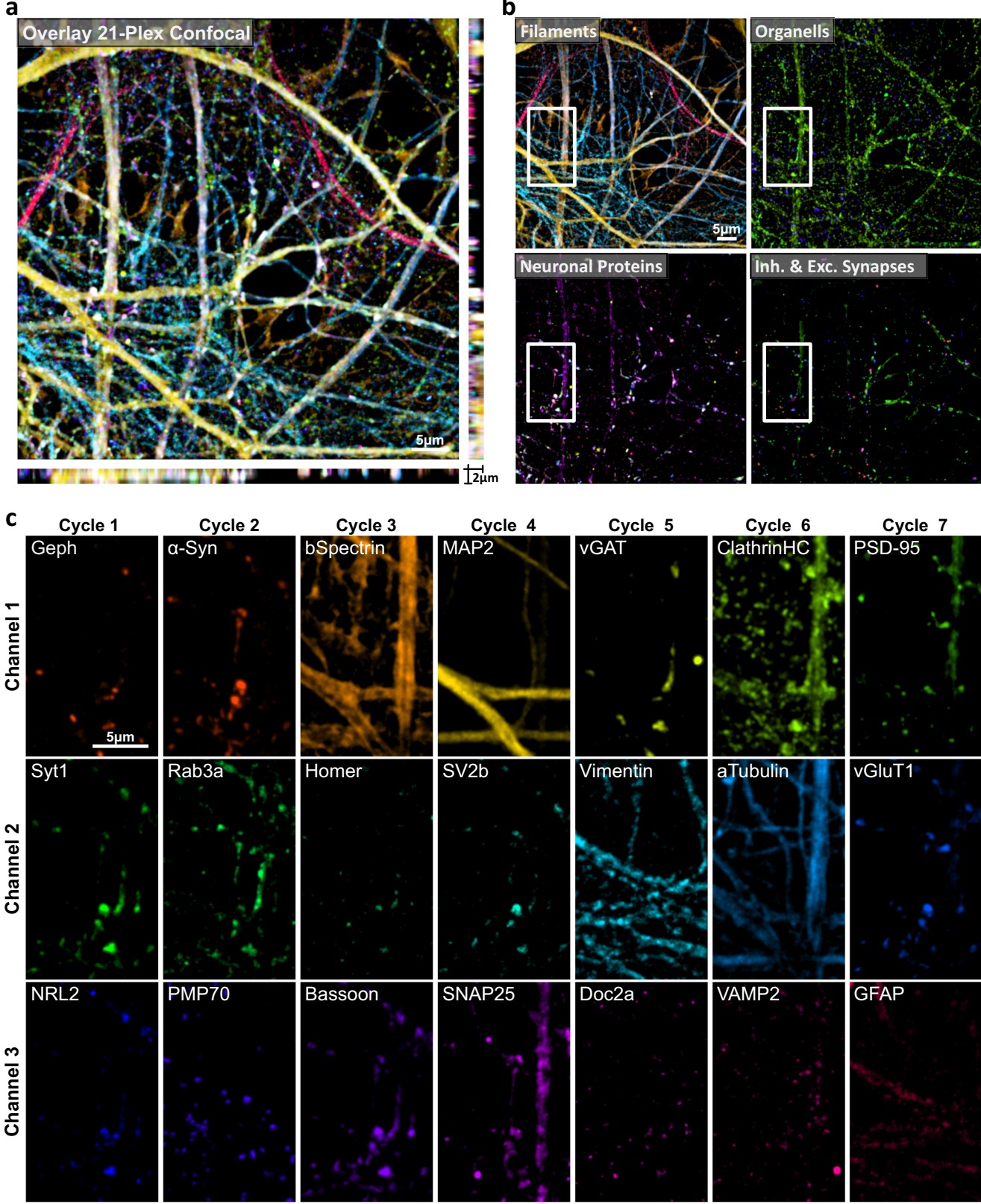

**Fig. 5 | Revealing 21 proteins on primary hippocampal neurons by ChemiPlex.** **a** Volumetric confocal image acquired after 7 repetitive cycles of ChemiPlex using 3-channel per cycle. **b** Categories of imaged targets comprised of 5 filaments, 3 organelles, 7 neuronal proteins, and 6 pre- and post-synaptic proteins. **c** High magnification images from individual proteins acquired during 21-ChemiPlex. From cycle 1 to 7 the following targets were revealed: Gephyrin (Geph), Synaptotagmin 1 (Syt1), Neuroligin 2 (NRL2), alpha-Synuclein (a-Syn), Rab3a, PMP70, beta-Spectrin (bSpectrin), Homer, Bassoon, Microtubule Associated Protein 2 (MAP2), Synaptic Vesicle protein 2b (SV2b), SNARE Associated Protein 25(SNAP25), vGAT, Vimentin, Doc2a, Clathrin, alpha-Tubulin, Synaptobrevin 2 (VAMP2), Post Synaptic Density 95 (PSD-95), vGluT1 and GFAP. Atto643, AZDye568, and Atto488 were used for channels 1, 2, and 3, respectively. Details on antibodies and one-step IF are in Methods and Supplementary Table 8.

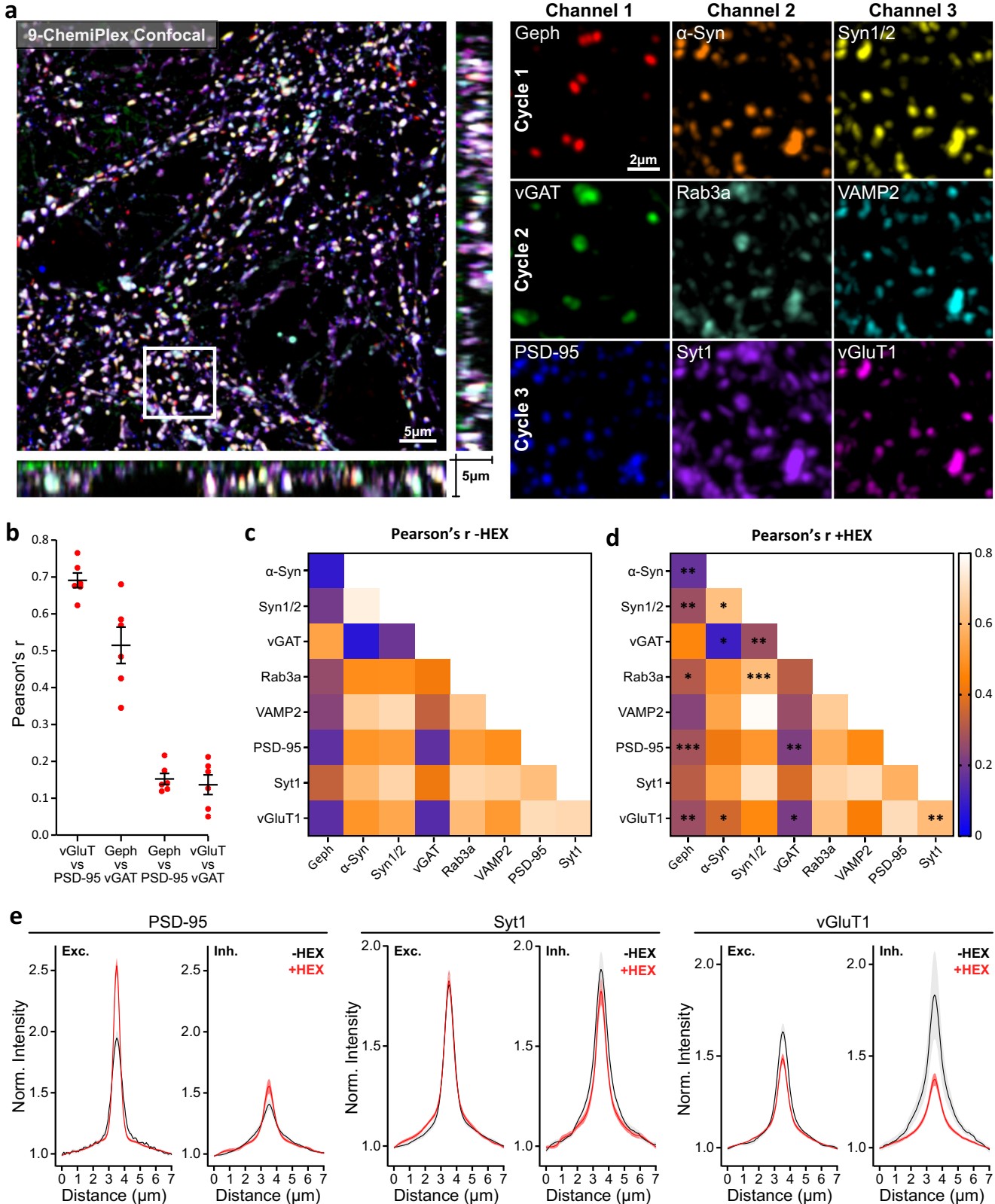

## Discussion

The complexity of cellular systems demands advanced methodologies, enabling the simultaneous visualization of multiple targets to observe and follow specific molecules in the context of other landmarks within a cell. Traditional methods, relying on indirect immunofluorescence (IF), have laid the foundation for multiplexed imaging by employing secondary antibodies (2.Abs) conjugated with distinct fluorophores to detect various primary antibodies (1.Abs). Despite some advantages offered by indirect IF, this method still encounters limitations when probing more than 4 or 5 targets using near Infrared emitting fluorophores[44] or up to 6 using spectral demixing approaches[45].

Addressing the need for a universally applicable and simpler approach for multi-target imaging, this study introduces NanoPlex, a

**Fig. 6 | Studying 9 synaptic proteins and the effect of 1.6-Hexanediol (HEX) using ChemiPlex. a** Volumetric confocal image acquired after 3 consecutive cycles of ChemiPlex using 3-channel per cycle. From cycles 1 to 3, the following targets were revealed on cycle 1: Gephyrin (Geph), alpha-Synuclein (α-Syn), Synapsin 1/2 (Syn1/2); on cycle 2: vGAT, Rab3a, VAMP2; and on cycle 3: Post Synaptic Protein 95 (PSD-95), Synaptotagmin 1 (Syt1) and vGluT1. Atto643, AZDye568, and Atto488 were used for imaging in channels 1, 2, and 3, respectively. **b** Pearson's correlation *r*-values from referenced pairs. The symbols show mean ± SEM. *n* = 6 regions from 3 independent coverslips. **c** A heat map of the Pearson's *r*-value. **d** Heat map of Pearson's *r*-value after treatment with HEX (+HEX). Statistical changes before and after treatment were determined by unpaired nonparametric Mann-Whitney tests and shown in (**d**). *$p < 0.05$, **$p < 0.03$, ***$p < 0.002$. *n* = 6 (−HEX), *n* = 12 (+HEX). Details on staining and Pearson's correlation distributions are in Supplementary Table 9 and Supplementary Fig. 12. **e** Fluorescence intensity variation of PSD-95, Syt1, and vGluT1 before (-HEX, black) and after (+HEX, red) treatment, measured along excitatory (Exc. PSD-95 positive) and inhibitory (Inh. gephyrin positive) regions of interest. The curves indicate mean ± SEM from *n* = 10 (−HEX), including 1573 gephyrin-positive and 5663 PSD95-positive synapses; and *n* = 12 for +HEX treatment, including 3148 gephyrin-positive and 7225 PSD95-positive synapses. Intensity distributions for the six remaining targets are in Supplementary Fig. 13.

versatile methodology leveraging engineered secondary nanobodies (2.Nbs) for iterative imaging. NanoPlex exploits the unique advantage of monoclonal secondary nanobodies (2.Nbs)[46], allowing a stoichiometric, controlled, and species-independent labeling of primary antibodies. The 2.Nbs designed for NanoPlex were functionalized with an ALFA-tag after the cleavable domains and were further revealed by a fluorescently labeled anti-ALFA nanobody[26]. This provided sufficient solubility to wash out the cleaved fluorophores efficiently and added flexibility when designing the experiments and matching the different 2.Nbs (i.e., anti-mouse IgG1 or IgG2 and anti-Rabbit IgG) and the needed fluorophores. Unlike existing methods relying on harsh treatments compromising sample integrity, NanoPlex employs diverse mild cleavable options to selectively remove the fluorescence reporter from the 2.Nb, while preserving the ultrastructure of cells and proteins. NanoPlex includes OptoPlex utilizing light-induced cleavage; EnzyPlex employing cleavage by a specific protease, and ChemiPlex utilizing redox chemistry for signal erasure.

OptoPlex has a unique advantage since it can be applied in a confined region of the specimen (Fig. 1e). The created Light Responsive Tag (LRT) can be cleaved with a moderate excitation light power of ~60 mW/cm² from commercial 365 nm LEDs (e.g., ~mW range of 405 nm laser is routinely used in STORM[47]). Therefore, it is not only compatible with the laser lines used in biological imaging but also holds the potential to be used in living samples following extracellular epitopes. EnzyPlex implementation does not require chemical conjugation like in the case of OptoPlex and ChemiPlex since the ALFA tag and protease substrate (bdSUMO) can be encoded genetically and produced as fusion protein when expressing the 2.Nbs. We choose the protease bdSENP1 because it is highly efficient and specific, and it can be used under various buffer conditions[29], including a wide range of temperatures and salt concentrations. Interestingly, bdSENP1 prefers a reducing environment for its activity, which makes it ideal for dSTORM microscopy, where reducing agents are needed to help with the blinking of fluorophores. Cleaving under a flowing buffer maximizes the catalytic activity by continuously removing the product (cleaved ALFA-tag and its associated fluorescence). EnzyPlex also has the potential for living sample applications since the protease is highly specific, having no activity on endogenous proteins of most cells and tissues (Fig. 2). Some potential drawbacks are the need for substantial amounts of bdSENP1, which, due to its relatively high cost, can be preferably produced in the lab (e.g., ~30€ per cleavage using yeast Ulp1 protease from ThermoFischer). In addition, if the imaging step uses intense excitation light, the enzyme's substrate could get photo-modified to the point of hampering its recognition by SENP1 and its efficient cleavage.

To overcome some of the EnzyPlex's limitations, we developed ChemiPlex, which employs 2.Nbs functionalized with a small cleavable domain (disulfide bond) that is not damaged during imaging and can be cleaved with a small, highly soluble, and commercially available reducing agent. For this, we choose Tris(2-carboxyethyl) phosphine (TCEP) due to its high solubility and stability in aqueous solution, providing reliable performances and no changes in pH while reducing as other reducing chemicals. TCEP is also very specific, targeting only disulfide bonds by donating electrons and losing its ability to reduce a second disulfide bond, working stoichiometrically, minimizing the undesired effects of a catalytic reducing agent. ChemiPlex provides a straightforward and effective strategy for signal removal across multiple targets, facilitating iterative imaging cycles without compromising the primary hippocampal neurons' ultrastructure or affecting epitopes or proteins (Fig. 5). It is important to notice that signal removal was not only different for each of the modalities. We think OptoPlex is the least efficient because the strong illumination needed for cleavage it also does some photochemistry fixing the fluorophores on the fluorescent NbALFA, making removal of signal less optimal. Interestingly, for EnzyPlex and ChemiPlex, efficiencies were very good, but we noticed that some targets had more difficulties than others for efficient cleaving, probably due to the target abundancy and direct chemical environment (luminal, in a large complex, membrane-associated, etc). Interestingly, abundant filamentous structures like microtubules or vimentin typically provide high signal removal.

Over the past years, several innovative approaches have emerged, ranging from iterative antibody stripping methods employing chemical denaturation[7] to sophisticated fluorescence lifetime methods[48–50] to more complex concepts like Exchange-PAINT and the recent Thermal-plex leveraging DNA barcoding to achieve multi-target imaging[21,51,52]. Moreover, these techniques were applied in combination and with special fluorophores, resulting in the imaging of 4 to 6 targets in diffraction-unlimited microscopy[53,54]. While these methods exhibit multiplexing capabilities, some only work with pre-defined targets (e.g., DNA revealed by Hoechs, filamentous actin by phalloidin), dedicated fluorophores, genetically manipulated POI (e.g., HaloTag), or under harsh treatments such as strong detergents, high temperature or the combination, hindering their widespread applicability and limiting the flexibility needed in biological studies. In contrast, Nano-Plex´s fundamental flexibility is that it can be used with any fluorophore and the lab's preferred antibodies, even when sourced from the same species, ensuring no compromise is needed when selecting the best-performing antibodies. In addition, the use of 2.Nbs eliminates the random conjugation when using NHS-ester chemistry on 1.Abs, which impacts flexibility and control over numbers and positions of the fluorophores or ssDNAs on the 1.Abs, also risking the loss of binding or its target-specificity.

An exception is a newly developed method termed SUM-PAINT (Secondary label-based Unlimited Multiplexed DNA-PAINT)[23], which enables high-multiplexing super-resolution using ssDNA as specific barcodes. This technique also takes advantage of the unique features of 2.Nbs and performs one-step immunostainings combined with the barcoding capability of PAINT. In addition, a similar method termed FLASH-PAINT was recently established using ssDNA directly conjugated on 1.Abs followed by ssDNA adaptors, achieving 13 Plex with a few nanometer precision[24]. However, most DNA-based approaches are relatively challenging to implement. They require thorough validations of each imager DNA to render specific signals with minimal background, and they require 10k–60k images per color/channel. This complexity leads to difficulties for non-specialized laboratories (e.g., requiring special hardware and expertise, as well as tailored imaging post-processing).

NanoPlex can be implemented with already available antibodies and microscopes in any laboratory, allowing the acquisition of high-content data without major effort. The applicability of NanoPlex was demonstrated across various microscopy techniques, including brightfield, confocal, dSTORM, and STED, showcasing its versatility and potential for universal multiplexing (Figs. 3 and 4). The robustness of NanoPlex was exemplified through multi-target imaging in primary hippocampal neurons, revealing intricate synaptic protein interactions and their response to perturbations induced by 1,6-hexanediol treatment (Figs. 5 and 6). With only 3 sequential cycles of ChemiPlex, we achieved straightforward confocal imaging of 9 synaptic proteins within primary neuronal cultures to showcase the co-localization of different pre- and post-synaptic markers, including both inhibitory and excitatory synapses. Determining the colocalization solely through indirect experiments might make it challenging, leading to uncertainty about whether the differences are due to sample variations or variations in how the stainings were performed in serial repetitions.

In conclusion, NanoPlex makes multiplexing universally accessible, allowing unlimited multi-target detection in any antibody-based assay, from microscopy to Western blot. Automatization should allow the unsupervised application of this method, which, if combined with recent advances in light-microscopy techniques delivering molecular resolutions (e.g., MINFLUX[55], RESI[35], ONE[43]), we expect this method to help the field of single-cell proteomics by moving closer to obtaining precise, and quantitative topological information of dozens or even hundreds of proteins within a single cell.

## Methods

This work complied with all biosafety, chemical and general security, and ethical regulations imposed by the institutions involved and the Lower Saxony state in Germany. Wild-type Wistar rats (*Rattus norvegicus*) were obtained from the University Medical Center Göttingen and were handled according to the specifications of the University of Göttingen and of the local authority, the State of Lower Saxony State Office for Consumer Protection and Food Safety (Niedersächsisches Landesamt für Verbraucherschutz und Lebensmittelsicherheit). Cultures were performed according to the ARRIVE guidelines (https://arriveguidelines.org).

### Chromatography

Thin-layer-chromatography was performed on 0.25 mm Silica-Gel 60 F plates with a 254 nm fluorescence indicator from MERCK. The substance detection took place by light with 254 nm and 360 nm wavelengths. Non-UV-active substances have been visualized by staining the plates with a potassium permanganate solution and gentle heating afterward. For the column chromatography, Merck Silica gel of the type Geduran® Si 60 (40–63 µm and 63–200 µm, 70–230 mesh ASTM) or Merck aluminum oxide standardized (according to Brockmann) was used.

### Melting points

The melting point of the synthesized LRT used in OptoPlex was determined with a Stuart® Melting Point Apparatus SMP10 from BARLOWORLD SCIENTIFIC.

### Nuclear magnetic resonance spectroscopy (NMR)

Recordings of NMR spectra have been performed on a BRUKER Avance III HD 300 at frequencies of 300 MHz (1H-NMR) and 75 MHz (13C-NMR). The chemical shift δ is shown in ppm, based on the standard tetramethyl silane. The solvent signal was used as an internal reference signal. The coupling constant J is shown in Hertz (Hz). For characterization, the following abbreviations have been used: s (singlet), d (doublet), and m (multiplett). The solvent signals are shown in Supplementary Table 1. The NMR-spectra was evaluated with MestReNova 14.2.0-26256 from Mestrelab Research S.L.

### Mass spectrometry (MS)

Mass spectrometry for the characterization of molecules and the confirmation of their identity was conducted via electrospray ionization (ESI) mass spectra and high-resolution ESI (HR-MS) spectra were recorded at a maXis or MicroTOF spectrometer by BRUKER DALTONIK GMBH (Bremen, Germany). The data were analyzed with Compass Data analysis software (version 4.0) by Bruker. One sample per compound was collected, and the samples were dissolved in MilliQ, MilliQ/MeCN mixtures, or MeCN. The values are given in m/z ratio, along with the relative intensity of the peak.

### Infrared spectroscopy (IR)

The infrared spectra have been recorded on a spectrometer from BRUKER of the type Alpha-P ATR. Liquid samples were measured as film, and solid samples were placed as pure substances. The evaluation of the spectra was done with Opus 6.5 from BRUKER. Therefore, the spectral range from 4000 cm$^{-1}$ to 400 cm$^{-1}$ was recorded.

### UV/VIS-spectroscopy and irradiation experiments

UV/VIS spectra were recorded on a SPECORD 600 spectrometer. Therefore, a spectral range between 200 nm and 800 nm has been recorded. Quartz cuvettes from HELLMA with a layer thickness of 1 cm were used. The evaluation was performed with Spectragryph v1.2.15 and OriginPro 2020. For irradiation, LEDs in a custom-built setup from Mountain Photonics (365 nm and 405 nm)[56] or from ThorLabs (445 nm, 505 nm, 625 nm) were used.

### High-performance liquid chromatography (HPLC)

Semi-preparative reverse-phase HPLC purifications were performed with a system from JASCO (Tokyo, Japan), consisting of two pumps PU-2020Plus, a 3-line degasser DG2080-53, and a diode array detector MD-2010Plus. Crude compounds were eluted with a linear gradient of two phases: phase A (MilliQ + 0.1% TFA) and phase B (MeCN + 0.1% TFA). An MN Nucleodor 100-5-C18, 250 mm × 10 mm, 5 µm column from MACHEREY-NAGEL (Düren, Germany) was used for purification, with a flow rate of 3 mL/min. UV detection was measured at 215 and 254 nm for non-labeled peptides. For LRT, absorbance was also monitored at 365 nm. For the preparation of the samples, they were dissolved in a 70:30 solution of MilliQ/MeCN, followed by filtration through a Chromafil filter from MACHEREY-NAGEL.

### Synthesis of light responsive tag (LRT)

As a photolabile core of the LRT, we positioned the photolabile bond between an ONB and a phenolether. Previous works have shown that phenolethers undergo a photo-cleavage at high rates[57], therefore, we choose it as the leaving group and linker between ONB and the maleimide handle and designed molecule **9** (Supplementary Fig. 1). An overview of the synthesis is given in Supplementary Fig. 1. The sequence starts with the preparation of vanillin derivative **1** following literature-known procedures[58,59]. The compound was then alkylated to introduce the alkyne and gave **2** in good yield (65%). Next, the benzaldehyde was converted reductively into the corresponding alcohol **3**, which was subjected to an Appel reaction to give bromide **4** (83%). Upon treatment with 4-hydroxybenzyl alcohol and potassium carbonate as base, the corresponding phenolether **6** was formed (quant.). The hydroxy group of the benzyl alcohol was brominated with phosphorus tribromide in a good yield (65%) to result in **7**. The benzyl bromide **7** was then converted to the maleimide **9** (13%), by the reaction with **8**[60], followed by a retro Diels-Alder reaction. Next, azido acetic acid (**10**) was clicked to the photolabile linker **9** in a copper-catalyzed azide-alkyne coupling (CuAAC). The mild, bioorthogonal conditions allowed the conversion of the molecule effectively in the presence of all functional groups. The same reaction was also attempted directly on azido-functionalized ALFA, however, was not successful in synthetically useful yields. We assume a strong mismatch

in the solubility of the peptide and the photolabile linker in solvents compatible with CuAAC being the reason for this. In contrast, **11** could be obtained successfully and was employed to synthesize the LRT. We employed microwave-assisted, automated solid-phase peptide synthesis (MA-SPPS) to obtain ALFA with a short, N-terminal GSG-linker (Supplementary Fig. 1c). Fmoc-deprotection and subsequent coupling with **11** resulted in LTR. The latter was purified using HPLC, and its purity and identity were confirmed with UHPLC and MS analysis.

Finally, the LRT was synthesized via manual SPPS. 0.01 mmol of resin-bound ALFA peptide was added to a BD syringe, and the resin was pre-swollen for 30 min, shaking in NMP. Deprotection was carried out twice by filling the syringe with 2 mL of a solution of 20% piperidine in NMP and shaking it at room temperature for 10 min. The supernatant was removed, and the resin-bound peptide was washed with NMP/CH$_2$Cl$_2$/NMP (3x). Afterward, compound **11** (26 mg, 0.05 mmol, 5 eq), DIC (8 μL, 0.05 mmol, 5 eq), and HOBt (7 mg, 0.05 mmol, 5 eq) were dissolved in 1 mL of NMP and added to the resin-filled syringe. The mixture was left shaking at room temperature for 2 h. The supernatant was removed, and the resin-bound peptide was washed with NMP/CH$_2$Cl$_2$/NMP (3x) and Et$_2$O (2x). The resin was dried in vacuum. Cleavage was performed according to SOP2, and the crude peptide was then purified via RP-HPLC (30 to 70% MeCN (+0.1% TFA) in MilliQ (+0.1% TFA) gradient over 25 min). A small aliquot of the pure peptide was collected for ESI-MS and UHPLC analysis. **HR-MS** (**ESI**+) – m/z calculated for [C105H160N34O37]: 2489.2; found: m/z 2489.2 [M + H]+; m/z 1245.5926 [M + 2H]2+; m/z 830.7308 [M + 3H]3+; m/z 623.3001 [M + 4H]4+.

## Cell culture

U2OS cells expressing endogenous nuclear pore complex protein Nup96 tagged with mEGFP (U2OS-Nup96-GFP)[28] were maintained in Petri dishes at 37 °C, 5% CO$_2$ in a humidified incubator using complete Dulbecco's Modified Eagle Medium (DMEM, ThermoFisher Scientific) supplemented with 10% Fetal Bovine Serum (FBS, ThermoFisher), 4mM L-glutamine and 1% pen/strep (ThermoFisher). For sample preparation, U2OS-Nup96-GFP cells were seeded on 18 mm poly-L-lysine-coated coverslips in 12-well plates and incubated for ca. 16 h at 37 °C, 5% CO$_2$ in a humidified incubator. Primary hippocampal neuron cultures were prepared as described[61]. Wild-type Wistar rats (*R. norvegicus*) were obtained from the University Medical Center Göttingen and were handled according to the specifications of the University of Göttingen and of the local authority, the State of Lower Saxony State Office for Consumer Protection and Food Safety (Niedersächsisches Landesamt für Verbraucherschutz und Lebensmittelsicherheit). Cultures were performed according to the ARRIVE guidelines (https://arriveguidelines.org). In brief, brains from P1-2 rat pups were extracted and placed in cold HBSS, and hippocampi were extracted. The hippocampi were triturated using a 10 mL pipette in complete-neurobasal medium (Neurobasal A, ThermoFisher), containing 2% B27 (ThermoFisher) and 1% Glutamax-I (ThermoFisher). Neurons were plated in 12-well plates on 18 mm glass coverslips coated with poly-L-lysin-hydrochloride (Sigma-Aldrich), in a plating medium (500 mL MEM, 50 mL horse serum, 5 mL glutamine, 330 mg glucose). After 2 h, the plating medium was replaced with a 1.25 mL complete-neurobasal medium, and neurons were incubated for 14 days in vitro (DIV) at 37 °C, 5% CO$_2$ in a humidified incubator. In Figs. 5 and 6, mature primary hippocampal neurons (DIV 14) were treated with 5% 1,6-hexanediol (#240117-50 G, Aldrich) for 5 min in the incubator. Cells were rapidly rinsed with pre-warmed Tyrode's solution (124 mM NaCl, 2.7 mM KCl, 10 mM Na$_2$HPO$_4$, 2 mM KH$_2$PO$_4$, pH 7.3) and fixed for further processing for immunolabeling.

## Cleavable secondary nanobodies

SdAb anti-Mouse IgG1, sdAb anti-Mouse IgG2a/b, and sdAb anti-rabbit IgG containing an ectopic cysteine on their C-terminal (NanoTag

Biotechnologies; Cat#: N2005, N2705, N2405) (Supplementary Table 11) were used. For ChemiPlex, these nanobodies were conjugated to a succinimidyl 3-(2-pyridyldithio)propionate (SPDP) ALFA-tag peptide (SPDP-ALFA-tag; custom-made by NanoTag Biotechnologies, Cat# N2007, N2707 & N2407, respectively). For the conjugation reaction, 30nmoles of each nanobody were reduced on ice using 10 mM TCEP solution at pH 6.5 for 60 min. Excess of TCEP was removed using Amicon spin filters with 10 kDa molecular weight cutoff (MWCO) (Merck, Cat#UFC500324). Immediately, 150nmoles of SPDP-ALFA-tag peptide were added and left overnight at 4 °C for disulfide exchange. SPDP-ALFA-tag conjugated nanobodies were further purified using a size exclusion chromatography column (Superdex® Increase 75, Cytiva) on an Äkta pure 25 system (Cytiva). For the OptoPlex approach, the same secondary nanobodies containing an ectopic cysteine on their C-terminal (NanoTag; N2005, N2705, N2405) (Supplementary Table 11) were conjugated to LRT, composed of a maleimide functionalized photocleavable o-Nitrobenzyl group followed by an ALFA-tag (see description below for the synthesis of this group). For coupling, nanobodies were reduced as above and incubated with ten molar excess of LRT; the reaction was left for 60 min at RT and overnight at 4 °C. The unreacted photocleavable peptide was removed using Amicon spin filters, 10 kDa MWCO. Secondary nanobodies for EnzyPlex were custom-produced from NanoTag Biotechnologies. Nanobodies were fused on their C terminus to a bdSUMO[29] domain followed by an ALFA-tag. All modified nanobodies were stored at −80 °C in a 50% Glycerol solution.

## bdSENP1 expression and purification

HisTag-TEV-bdSENP1 was expressed in the NEB Express bacteria strain. Bacteria were grown on terrific broth supplemented with kanamycin at 30 °C. When OD600 reached ~3, 0.4 mM IPTG was added. Expression was induced overnight (-16 h) under shaking. Bacteria were pelleted and resuspended in cold lysate buffer (LysB: 100 mM HEPES, 500 mM NaCl, 25 mM imidazole, 2.5 mM MgCl$_2$, 10% v/v glycerol, 1 mM DTT, pH 8.0) supplemented with DNAse, lysozyme and 1 mM PMSF. After 30 min of incubation and disruption by sonication (Brenson Inc.), the lysate was centrifuged at -11,000 × g for 1 h at 4 °C. The supernatant was incubated with LysB equilibrate Ni+ beads (Roche cOmplete Resin) for 1 h at RT. Beads were washed with 3 column volumes using LysB buffer, 5 CV with high salt buffer (50 mM HEPES, 1.5 M NaCl, 25 mM imidazole, 2.5 mM MgCl$_2$, 5% v/v glycerol, 1 mM DTT, pH 7.5). Finally, before the elution, beads were washed in the buffer of choice for the next application. Elution was carried out using LysB containing 400 mM imidazole. Fractions were incubated overnight with TEV protease and, the next day passed through a size exclusion chromatography (Superdex 75-Increase, ÄKTA 25pure, Cytiva). bdSENP1 was finally cleaned of HisTag-TEV by using reverse-nickel purification. Purity and activity were evaluated in SDS-PAGE.

## SDS-PAGE cleavage assessment

120 nM of each 2.Nb fused to the bdSUMO-ALFA-tag domain was incubated with or without 1 μM of SENP1 protease for 10 min. Samples were then mixed with a 2x Laemmli buffer, boiled at 96 °C for 10 min, and loaded in 12% SDS-PAGEs prepared using the Bio-Rad system. Gels were then stained using Coomassie Blue. Gels were imaged using the Amersham Imager 600.

## Immunofluorescence

U2OS-Nup96-GFP cells and primary neurons imaged under epifluorescence or laser scanning confocal microscopy were fixed with pre-warmed 4% paraformaldehyde (PFA) for 30 min at RT. For STED and dSTORM microscopy, U2OS-Nup96-GFP cells were first treated for 30 s with pre-warmed Extraction Buffer (EB, 10 mM MES, 138 mM KCl, 3 mM MgCl$_2$, 2 mM EGTA, 320 mM sucrose, pH 6.8) supplemented with 0.1% Saponin. This was directly followed by fixation with a pre-

warmed solution of 4% paraformaldehyde and 0,1% Glutaraldehyde in EB for 15 min at RT. After removing the fixatives and rinsing with PBS, unreacted aldehyde groups were quenched using 0.1 M glycine in PBS for 15 min. Cells were blocked and permeabilized using 3% bovine serum albumin (BSA) and 0.25% Triton X-100 (Sigma Aldrich) in PBS for 30 min at RT. Each individual one-step immunofluorescence was set by pre-mixing 1.Abs, with cleavable 2.Nbs, and fluorescently labeled NbALFA in a 1:3:4 molar ratio (see Supplementary Tables 10, 11). Alternatively, directly-labeled, no cleavable 2.Nbs, or primary nano-bodies were used (Supplementary Table 11). Each pre-mixture was initially done in 20 µl PBS and left at RT for 30 min to form stable complexes. Then, the formed complexes were diluted to the intended final concentration of 1.Abs and added to the sample for 60 min at RT in 3% BSA, 0.25% Triton X-100. Samples were washed thrice with PBS for 5 min each and mounted on a custom-made flow-chamber (Supplementary Fig. 4b). The chamber was filled with Washing-Buffer (0.1 M Glycine in PBS, Supplementary Table 2) and plugged into a two-way closed fluidics system connected to a peristaltic pump (MINIPULS 3 Pump, Gilson).

## Cleaving efficiency

The fluorescence drops over time was monitored employing laser scanning confocal microscopy to image immunolabeled U2OS-Nup96-GFP cells before and after applying the cleaving buffers (Supplementary Fig. 2) or light. To track OptoPlex, EnzyPlex and ChemiPlex cleaving, samples were immunolabeled with pre-formed cleavable fluorescence complexes of anti-vimentin mouse antibody (10 nM) (Santa Cruz, sc-6260), secondary anti-mouse IgG1 OptoPlex nanobody (30 nM), and FluoTag®-X2 anti-ALFA coupled to Atto643 (40 nM). All samples were mounted on custom made chamber for fluidics (Supplementary Fig. 4b). Buffers (see Supplementary Table 2) were flushed at 1 ml/min, and flow was stopped during image acquisition. Photo-cleaving was monitored before and after continuous application of LED light (365 nm, ~60 mW/cm²) for 15 min. Bleaching control experiments used the same fluidics setup and image acquisition settings according to the tested cleavable nanobody. For bleaching controls, staining used the same primary antibody, pre-mixed with directly labeled secondary nanobodies instead of the cleavable secondary nanobodies (using the same fluorophore). Bleaching controls were performed using secondary FluoTag®-X2 anti-Mouse IgG1 directly coupled to Atto643 (NanoTag; N2402-At643) or Atto488 (30 nM) (NanoTag; N2002-At488).

## NanoPlex

Repetitive cycles of single-step immunolabeling (Step 1), imaging (Step 2), signal erasing (Step 3), and inactivation (Step 4) were carried out to generate 3D confocal images from 6 different proteins in U2OS-Nup96-GFP cells using OptoPlex (Fig. 1), EnzyPlex (Fig. 2) and Chemi-Plex (Fig. 3). Information for the probes pre-mixtures used in OptoPlex, EnzyPlex, and EnzyPlex 3D 6-Plex can be found in Supplementary Tables 3, 4, 6, respectively. Using a similar approach, STED images from 8 different subcellular structures in U2OS-Nup96-GFP cells (Fig. 4), and 3D confocal images from 9 (Fig. 6) and 21 (Fig. 5) different proteins in primary neuronal cultures were acquired employing Che-miPlex. *ChemiPlex*: For 8-Plex STED microscopy, U2OS-Nup96-GFP cells were initially labeled against mitochondria with pre-formed complexes of primary anti-TOM20 antibody (see Supplementary Tables 10, 11 for all antibody details), and each single-step immunola-beling was performed as indicated in Supplementary Tables 6–9. The coverslips were mounted on the imaging chamber (Supplementary Fig. 4), filled with the washing buffer, and plugged into the fluidics system. All buffers were pumped at 1 ml/min using a peristaltic pump. The washing buffer was replaced with the imaging buffer (see buffers on Supplementary Table 2), and flow was stopped during the acquisition of the STED images (Step 2, Fig. 1a). After imaging, the washing

buffer flowed for 2 min before being replaced with a ChemiCleave buffer containing the reducing agent (10 mM TCEP, Step 3). The cleaving buffer was applied for 15 min. Reduced (reactive) thiol groups were inactivated (Step 4) by flowing through the sample a thiol-quenching buffer (20 mM NEM, 0.2 M NaCl, in PBS) for 5 min. Finally, the imaging buffer was again introduced, and a post-erasing STED image was acquired. After control images were done, the imaging buffer was washed out, and the next single-step immunolabeling was introduced to initiate a new imaging cycle. For laser scanning confocal microscopy, confocal z-stacks (3D) imaging in primary neurons were carried out as described above for 8-Plex-STED imaging. However, for this modality, three targets were labeled and imaged on each cycle; thus, only 3 or 7 cycles were needed to image 9 and 21 targets. For each cycle, three single-step immunolabelings were performed. Each pre-mixture of complexes was performed individually in different tubes, and each pre-formed complex carried a different fluorophore (Atto488, AzDye568 & Atto643, See Supplementary Tables 8 and 9). The three formed complexes were pulled together, including 100 nM of unlabeled secondary nanobodies (NanoTag; K0102-50, K0202-50) to block potential crosstalk between the secondary probes and incu-bated with the sample simultaneously. Supplementary Table 8 shows all of the pre-formed complexes employed for the immunolabeling of 21 neuronal targets (Fig. 5) and Supplementary Table 9 shows the preformed complexes used in 9 neuronal targets imaging (Fig. 6). *EnzyPlex:* Labeling and imaging using 3D confocal imaging (Fig. 2) were performed as described for ChemiPlex, with the difference that cleaving occurred by adding a buffer containing the specific SENP1 protease and not a reducing agent, resulting in the removal of the fluorescent signal. Inactivation (Step 4) was achieved with the same buffer as used in ChemiPlex (20 mM NEM, 0.2 M NaCl, in PBS; Supplementary Table 2) since NEM also inactivates potential traces of SENP1 protease, allowing it to proceed with a new labeling cycle. The same procedure was used to obtain dSTORM images from 5 different targets upon employing 5 repetitive cycles of EnzyPlex. For this, we used NbALFA conjugated to JaneliaFluor635b (JF635b), and the ima-ging was performed in PBS. Supplementary Tables 4 and 5 show all of the pre-formed complexes employed for EnzyPlex. *OptoPlex:* U2OS-Nup96-GFP cells were single-step immunolabeled using the same strategy as Chemi- and EnzyPlex but with the photocleavable-capable secondary nanobodies. Cleavage illumination was achieved using a 365 nm LED at ~60 mW/cm², measured at the sample using a power and energy meter interface with USB operation (PM100USB, Thorlabs, Inc.) along with a microscope slide power sensor (S170C, Thorlabs, Inc.). Cleavage was performed by continuous illumination for 15 min while flowing the washing buffer (see Buffer Supplementary Table 2). Supplementary Table 3 shows all of the pre-formed complexes employed for OptoPlex (Fig. 1).

## Epifluorescence microscopy

For Fig. 1e and Supplementary Fig. 5a images from 400 neighboring regions were stitched together under the inverted epifluorescence Nikon Eclipse Ti-E microscope (Nikon Instruments Inc.) equipped with a Nikon DS-Qi2 camera, an HBO-100W lamp, and a 60x oil immersion objective. The illumination settings (ND filter of 16 and exposure for 200 ms and max gain of the camera) were configured using the NIS-Elements AR software, version 4.60.00 (Nikon Corporation), and kept constant for imaging the sample before and after photo-cleaving.

## Confocal and STED microscopy

Confocal and STED images of neurons and U2OS were acquired with an STED Expert line microscope (Abberior Instruments, Göttingen, Germany) composed of an IX83 inverted microscope (Olympus, Hamburg, Germany) with an UPLSAPO 100 × 1.4 NA oil immersion objective (Olympus). Additionally, a pre-installed Autofocus system (Abberior Instruments, Göttingen, Germany) compensated for z-drifts while 2D

images were acquired. For cleaving efficiency assessments, samples were imaged in the confocal mode under low laser exposure settings to avoid photocrosslinking and retention of the cleaved fluorescent groups. Thus, excitation lasers of 640 nm and 405 nm were each set at a power level of 1 μW, (measured at the sample stage with power and energy meter interface from PM100USB, Thorlabs, Inc. and microscope slide power sensor S170C, Thorlabs, Inc). The pixel size was set to 60 nm, the pixel dwelling time was 5 μs, and photons from each scanning line were accumulated only 1 time. Iterative volumetric imaging cycles of ChemiPlex in neurons were carried out with excitation lasers of 640 nm, 561 nm, 488 nm, and 405 nm. For EnzyPlex and PhotoPlex of U2OS cells, 640 nm, 561 nm, and 488 nm lasers were employed. The x−y pixel size was set to 70 nm, the z-pixel size was 500 nm, and the dwelling time was 7 μs with 2 lines accumulation of signal. STED microscopy was employed for iterative cycles of ChemiPlex where 640 nm excitation and 775 nm depletion lasers were used at 1 μW and 5 mW, respectively. The pixel size was set to 20 nm, the dwelling time was 7 μs, and photons from each scanning line were accumulated three times.

## dSTORM imaging

Wide-field measurements were performed using a custom-built optical setup, described before[62] and displayed in Supplementary Fig. 7. First, fiducial markers (90 nm gold nanoparticles, G90-100 Cytodiagnostics) were incubated for 5 min with the sample. The unbound gold nanoparticles were removed with a thorough wash with PBS pH 7.4. Then, an imaging buffer (Supplementary Table 2) was introduced to minimize photodamage during the movie acquisition. Afterward, cells suitable for imaging were detected using low laser power (-1 mW, 640 nm), and dSTORM imaging was performed with an order of magnitude increased laser power (-10 mW, corresponds to a power density of -0.5 kW/cm$^2$) to induce blinking of JF635b dye. A movie of ~40k frames was acquired with an exposure time of 30 ms per frame. During the acquisition process, the focus was maintained manually based on the shapes and widths of fiducial markers. All experiments were done at 23 °C to maintain the mechanical stability of the microscope. Repetitive cycles of EnzyPlex were performed to reveal tubulin, clathrin vesicles, vimentin, peroxisomes, and nuclear pore complexes (Fig. 2).

## dSTORM image analysis

All acquired images were processed and visualized using Fiji/ImageJ (v. 1.53o). To depict the data, high-frequency noise was reduced using a Fast Fourier Transform filter set at 2 pixels. Brightness and contrast were applied uniformly to all parts of the images, and line profiles were obtained along a 3-pixel width. For cleaving efficiency assessments, images before treatment were thresholded using the RenyiEntropy algorithm, and signal-specific regions of interest were created and saved for analysis. To define the background of each image, the signal-specific regions were removed, and a Gaussian filter with sigma = 3 was applied. Background images were created using an eroding ball of 50 pixels, and their max intensity values were used to threshold the initial image of interest. All pixels with values below the threshold were replaced with a non-applicable number (NaN). Signal-specific regions of interest were selected once more on the background corrected images, and the intensities were measured before and after treatment initiation. For dSTORM image reconstruction, the acquired movies were analyzed with an ImageJ plugin ThunderSTORM[63]. Single emitters were localized using an intensity threshold, and final super-resolution images were reconstructed. All movies were corrected for lateral drift based on fiducial marker positions, and out-of-focus localizations were filtered out by thresholding the width of single-molecule point spread function. Finally, super-resolution images of all targets were overlaid and aligned using the fiducial markers coordinates and the Fast4DReg plugin[64]. NeNa algorithm[31] was employed to calculate dSTORM

localization precision, as it provides a robust way to estimate the image quality by leveraging the distances between nearest neighbors. This metric is crucial for assessing the quality of our dSTORM images quantitatively. The average localization precision of the reconstructed super-resolution images was 12.9 nm.

## Colocalization analysis

For cleaving efficiency analysis, before and after treatment, images from the same ROI were aligned using the Fast4DReg plugin in the Fiji/ImageJ software. Images were corrected for background, and the fluorescence intensity of the structures of interest was measured. The line intensity profiles of the denoted regions in the images of Figs. 1h, 2e, and 3d were plotted for each protein of interest before and after treatment. For colocalization analysis, multiple images from the same ROI were aligned using the Fast4DReg plugin. Pearson´s correlation coefficients (Pearson's r-values) were calculated using the JaCOP plugin[65] in the Fiji/ImageJ software (Fig. 6, Supplementary Fig. 12).

## Statistics & reproducibility

All statistical analysis procedures took place in GraphPad Prism (v. 9.1.2) and are described in the legends of each Figure. Plotted data are shown as mean ± SD or mean ± SEM, and statistical comparisons were made by unpaired nonparametric Mann-Whitney tests, as indicated in the Figure legends. No statistical method was used to predetermine the sample size. No data were excluded from the analyses. The experiments were not randomized, and the Investigators were not blinded to allocation during experiments and outcome assessment.

## Reporting summary

Further information on research design is available in the Nature Portfolio Reporting Summary linked to this article.

## Data availability

The source data file provided with this paper contains all the data used in the work presented here. The lead contact will share raw images sets upon request. Source data are provided with this paper.

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

## Acknowledgements

We thank Luke Levis (hhmi, Janelia) for providing maleimide-functionalized JF635b. We also want to thank Jannik Hentze for his fantastic assistance in the laboratory. We thank Eugenio Fornasiero for helpful proofreading and providing comments on the manuscript. The authors also thank Dr. Samrat Basak for developing a protocol for handling and applying fiducial markers for EnzyPlex dSTORM imaging. S.O.R. and F.O. were supported by Deutsche Forschungsgemeinschaft (DFG) through the SFB1286 (project A03 and Z04, respectively). This work was funded by the Deutsche Forschungsgemeinschaft (DFG, German Research Foundation) under Germany's Excellence Strategy (EXC 2067/1-390729940 to NAS).

## Author contributions

N.M. performed and analyzed most of the experiments to establish NanoPlex. E.R.C., N.I. and N.A.S. designed the LRT molecule and performed its synthesis and photochemical characterization. R.T. contributed with a custom-built wide-field setup and performed dSTORM imaging and analysis. J.S. helped with Confocal & STED imaging. S.O.R. helped with analysis and data interpretation. F.O. conceived the concept, designed experiments, interpreted data, and supervised the study. N.M., N.A.S., R.T., S.O.R. and F.O. wrote the manuscript with contributions from all authors. All authors reviewed and approved the final manuscript.

## Funding

## Competing interests

F.O. and S.O.R. are shareholders of NanoTag Biotechnologies GmbH. All other authors declare no competing interests.
