## [Peer Review File · Nature Communications]

Reviewers' Comments:

Reviewer #1:

Remarks to the Author:

This study takes advantage of click chemistry to generate a versatile multiplexing secondary nanobody allowing iterative reimaging of the same samples reaching up to 21 targets for 3D confocal analyses and 5-8 targets for super-resolution imaging techniques such as dSTORM and STED. To enable re-iterative immunostaining, the authors have put together 3 molecular erasing systems to remove the previously imaged fluorescent signal based on light, enzymatic and chemical modifications. The authors went on to show altered co-localization of key proteins localised on synaptic vesicles before and following treatment with 1,6-Hexanediol. Since 1,6-Hex interferes with liquid-liquid phase separation, their results provides supporting evidence that their novel techniques can be used to accurately quantify subcellular re-distribution of proteins. The universality of this technique is appealing as it will allow labs to work with their preferred antibodies, regardless of species, ensuring flexibility and no compromise on antibody performance. This study is well carried out, easy to follow and therefore likely to generate significant interest. I only have a few issues.

Major Points:

- For the most part the manuscript is well written, but there are several instances where vital information is absent, and statements are made with little context or tangible quantification. The manuscript would benefit from some re-arrangement of data from the supplementary tables and figures into the main text. In its current formatting, no statistical comparisons or validation of the imaging methods for the 2.Nb labels, are within the main body of the text. The experimental pipeline, particularly for ChemiPlex/STED is complex and would need to be shown in illustrative format. The material and methods sections are missing information on microscope set up and image acquisition settings, nearest neighbour analysis, and the figure legends omit key information on the experimental paradigm, as they should be able to stand alone.

- Both Optoplex and EnzyPlex reported non-homogeneous cleavage efficiency for all targets. This is suggested to be due to inaccessibility of the bdSENP1 for EnzyPlex, or by limited diffusion within the samples altering imaging cycles for either approach. More discussion is needed to clarify which targets were not suitable to imaging using these approaches. By comparison, ChemiPlex was reported to result in a very efficient and homogenous cleave through five cycles, but all of these analyses are in stated to be with in the Supp Figs. No details in the main text (nor supp data) appear to discuss how efficiency was assessed statistically.

Minor points:

No context has been provided for why nearest neighbour (NeNa) analysis was

completed on the dSTORM dataset. NeNa can be used to assess localization precision, was the analysis used to assess that the precision of imaging was unaffected by iterative cycles of EnzyPlex? The full NeNa analysis should be included provided.

It is unclear if the ChemiPlex system is compatible with dSTORM imaging approaches. Since reducing agents are used in STORM imaging buffers (including TCEP), the authors should discuss why they validated their approach using STED imaging following ChemiPlex and not dSTORM (as used for the other two approaches).

More detail on the experimental pipeline is needed in each figure 1-3. Figure 1a, 2a and 3a do not provide sufficient detail of the steps involved in the process. Modifying or relocating Supp Fig 4 would be recommended.

Similarly, since ChemiPlex followed by STED imaging compromises the most significant part of the manuscript, a more detail explanation or illustrative figure would be needed to understand the full extent of the timeline and critical steps involved. The validation of the signal 'cleavage' is found in Supp Fig 8 and 9 and appears to be on a single set of images only. Some statistical justification of its efficiency is needed to be shown in the main text.

More generally, there is a heavy reliance upon important data being only available in the supplementary figures. Key methodological information is missing from Figures and Figure legends alike.

Methods lack detail: Many chemicals formula do not have subscript numbering formatted. Nearest neighbour analysis procedure is missing. Melting point procedure is lacking detail on what experiment it refers to. Buffer contents should be within methods text rather than just in Supplementary Table 2.

Figure 1h, 2e and 3d: pseudocolouring for the individual panels is very difficult to see. How the images were acquired or processed is absent from each of these figure legends, Are these z-projections?

Figure 1c, 2c and 3c should show the comparison of the GFP Normalised intensity profile (relocation from relevant Supp Figs).

Line 47-49: The logic of this sentence is unclear.

Line 123: U2OS-Nup96-GFP cells are not outlined in any detail. Why were they chosen?

Line 357 (and 366): EndNote formatting error.

Line 422-424: The following statement in the discussion has not been well justified or explained:

“EnzyPlex was designed for a more straightforward implementation since 2.Nbs can be expressed as fusion proteins, including the bdSENP1 protease substrate (bdSUMO), followed by an ALFA-tag.”

Line 595: Italicize species name.

Line 604: Typos. 18mm coverslips, not mM. Poly-L-Lysin'e'.

Line 654: Typo. Laemmlli.

Line443-449: These sentences need to be re-written as the message is unclear.

Line 753: Include image illumination settings.

Line 764-774: It is said that that pixel size is 'set' to a specific size. The technical detail on how the Confocal and STED imaging was performed is lacking.

Reviewer #2:

Remarks to the Author:

This article presents a set of 3 different strategies to obtain multiplex imaging using sequential immunostaining. All 3 methods are based on secondary nanobodies that are cleavable in different ways, and can therefore be removed from the target (without damaging the sample) before doing another round of immunostaining. One method is optical, using UV light with an efficiency of ~70%, another one is enzymatic, with an efficiency of 90% and the last one is chemical, using TCEP with again an efficiency of 90%. All three methods are demonstrated using different imaging modalities (confocal and and super-resolution STED and dSTORM) for a variety of different targets, and for different cell lines (U2OS and primary neurons). Moreover, quantitative analysis of the spatial correlation between different synaptic markers is demonstrated. This is an important development, as the increase in multiplexed imaging requires efficient staining mechanism. Compared to DNA-based approaches, which are very powerful for single molecule imaging, these nanoplex also allows epifluorescence and confocal imaging. The article is well written, the methods section is comprehensive, and a lot of technical information is provided in the supplementary material. I think this article should be published, but I do have a few questions:

1. Optoplex shows reduced removal efficiency compared to the other approaches, but is the easiest method to implement. I know time is typically not of the essence in these experiments, but have the authors tried using higher UV intensity? This would be particularly relevant for FRAP-like experiments on live samples, where waiting several minutes would be detrimental. Please also normalized the power of the 365nm light used by the surface illuminated to get the intensity in W/cm².
2. dSTORM imaging is demonstrated using Enzyplex, but using self-blinking dyes. Can the authors comment on the compatibility with common STORM buffers (GLOXCAT or PCAPCD+Thiol).

3. Same question but about Chemiplex: are the thiols typically used for dSTORM compatible with Chemiplex?

4. In Chemiplex, the authors modify the protocol between fig 4 and 5-6 and add unconjugated secondary nanobodies to reduce background signal, can they provide a quantification of how the background signal increases as a function of the number of cycles with and without this step?

Other comments:

- 4 color imaging is something I would consider standard, but increasing this to 5 colors (by using redder dyes, like cy7 for example) is straightforward. It would also be useful to mention in the introduction spectral-unmixing based approaches, which allow imaging of 6+fluorescent markers. [for example, Valm et al, "Applying systems level spectral imaging and analysis reveal the organelle interactome" Nature, 2017]

-Please add a reference to the newly published multiplexed DNA FASH-PAINT (Schueder et al. Cell 2024)

Typo: fix reference to {Mivolanovic2017} at the end of the results section.

Reviewer #3:

Remarks to the Author:

In this manuscript, Mougios et al. present a new approach for iterative immunofluorescence (IF), NanoPlex, which uses conventional primary antibodies combined with secondary nanobodies engineered for three distinct removal strategies: OptoPlex, which uses a photocleavable moiety; EnzyPlex, which uses an enzymatic cleavage; and ChemiPlex, which exploits redox chemistry to induce cleavage. While various methods have previously been developed (and are cited by the authors), NanoPlex avoids harsh chemical conditions that could disrupt sample integrity, is compatible with a range of imaging modalities (including super resolution), uses off-the-shelf primary antibodies, and offers flexible signal removal strategies that may enable multiplexed IF in specific contexts not currently served by available alternative strategies. This is an innovative approach that is a welcome addition to current multiplexing strategies. Given that the main weakness of this approach (as noted in the manuscript by the authors) is the extent of signal removal after cleavage, more extensive quantification of this residual signal is required. The authors should address the following points related to residual signal prior to publication:

The authors perform similar experiments to quantify the extent of signal removal over

time for each of their removal strategies in Figs 1g, 2d, and 3c. However, each of these experiments uses a different primary antibody for this experiment. It would be beneficial to the readers assessing these different approaches to see how the strategies compare for the same primary antibody.

The text reports that vimentin is used to assess EnzyPlex (pg 6, line 12); however, Fig 2c-d reports tubulin staining.

While representative images provide some indication of the extent of signal removal of each method across multiple antibodies, I would like to see a quantification of residual signal after cleavage for all multi-round experiments similar to what was performed in Fig 4C. I am specifically referring to the experiments presented in Fig 1h, Fig 2e, Fig 2f, and Fig 3d.

Could the authors comment on the accumulation of residual signal over successive imaging cycles? For example, in Supp Fig 9, the authors quantify residual signal in 8-ChemiPlex STED microscope after each cycle. Each cycle uses the same fluorophore, therefore residual signal from each round should remain in all subsequent cycles, if we assume that each cycle of TCEP reduction is performed to completion for each antibody (i.e., maximum signal removed). If the authors quantified the intensity of residual signal after each cycle using the same imaging parameters, they could report on the extent of signal accumulation over successive rounds. This experiment would not necessarily have to be performed using STED.

What is the contribution of residual signal to the colocalization analysis performed in Fig 6b-d? When comparing two proteins imaged using the same fluorophore (e.g., vGAT and gephyrin), residual signal from the first antibody should be present when imaging the second antibody and thus artifactually contribute to the quantification of colocalization. In this case (and in all other multi-cycle experiments performed in this manuscript), the authors could image the sample after cleavage and use this image to correct for residual (non-specific) signal from the subsequent cycle (after image registration to align the images). To calculate the contribution of residual signal to colocalization calculations, the authors should compare the Pearson's r in the uncorrected images presented in the manuscript for given antibody pairs versus images in which residual signal has been subtracted, as described above.

Reviewer #4:

Remarks to the Author:

I co-reviewed this manuscript with one of the reviewers who provided the listed reports.

This is part of the Nature Communications initiative to facilitate training in peer review and to provide appropriate recognition for Early Career Researchers who co-review manuscripts.

Replies to the Reviewer Comments

Please find below our replies to all of the comments of Referees.
The text of our replies is colored in blue.

Reviewer #1 (Remarks to the Author):

This study takes advantage of click chemistry to generate a versatile multiplexing secondary nanobody allowing iterative reimaging of the same samples reaching up to 21 targets for 3D confocal analyses and 5-8 targets for super-resolution imaging techniques such as dSTORM and STED. To enable re-iterative immunostaining, the authors have put together 3 molecular erasing systems to remove the previously imaged fluorescent signal based on light, enzymatic and chemical modifications. The authors went on to show altered co-localization of key proteins localised on synaptic vesicles before and following treatment with 1,6-Hexanediol. Since 1,6-Hex interferes with liquid-liquid phase separation, their results provides supporting evidence that their novel techniques can be used to accurately quantify subcellular re-distribution of proteins. The universality of this technique is appealing as it will allow labs to work with their preferred antibodies, regardless of species, ensuring flexibility and no compromise on antibody performance. This study is well carried out, easy to follow and therefore likely to generate significant interest. I only have a few issues.

We thank the reviewer for the positive assessment of our work.

Major Points:

- For the most part the manuscript is well written, but there are several instances where vital information is absent, and statements are made with little context or tangible quantification. The manuscript would benefit from some re-arrangement of data from the supplementary tables and figures into the main text. In its current formatting, no statistical comparisons or validation of the imaging methods for the 2.Nb labels, are within the main body of the text. The experimental pipeline, particularly for ChemiPlex/STED is complex and would need to be shown in illustrative format. The material and methods sections are missing information on microscope set up and image acquisition settings, nearest neighbour analysis, and the figure legends omit key information on the experimental paradigm, as they should be able to stand alone.

We have made more quantification and statistics of signal removal and added to the main Figures. We have worked on clarification of the procedures in general and added a graphical description in Supp. Fig. 9 regarding the steps for ChemiPlex under the STED regime. We have improved and added substantially more details on Figure legends and Material and Methods, including explanations for image analysis, such as nearest neighbor analysis NeNa.

- Both Optoplex and EnzyPlex reported non-homogeneous cleavage efficiency for all targets. This is suggested to be due to inaccessibility of the bdSENP1 for EnzyPlex, or by limited diffusion within the samples altering imaging cycles for either approach. More discussion is needed to clarify which targets were not suitable to imaging using these approaches. By comparison, ChemiPlex was reported to result in a very efficient and homogenous cleave through five cycles, but all of these analyses are in stated to be with in the Supp Figs. No details in the main text (nor supp data) appear to discuss how efficiency was assessed statistically.

We brought from Supp. figures and added new quantification and statistics of signal removal efficiency for all modalities (OptoPlex, EnzyPlex, and ChemiPlex) and further discussed potential reasons why erasing signals for some specific targets seems to be systematically poorer than others in the Discussion section.

Minor points:

No context has been provided for why nearest neighbour (NeNa) analysis was completed on the dSTORM dataset. NeNa can be used to assess localization precision, was the analysis used to assess that the precision of imaging was unaffected by iterative cycles of EnzyPlex? The full NeNa analysis should be included provided.

Yes, NeNa was used to assess the precision obtained in every cycle during dSTORM imaging, showcasing that all cycles resulted in a super-resolved image with good localization precision. This is a routine analysis done in the field. We added more context and clarification on the main text and extra clarification on how NeNa was determined.

It is unclear if the ChemiPlex system is compatible with dSTORM imaging approaches. Since reducing agents are used in STORM imaging buffers (including TCEP), the authors should discuss why they validated their approach using STED imaging following ChemiPlex and not dSTORM (as used for the other two approaches).

No, ChemiPlex won't be compatible with conventional dSTORM, which needs reducing agents in the imaging buffer to promote the blinking of fluorophore; this makes ChemiPlex incompatible with conventional dSTORM. We have made this point clearer in the main text. We chose STED to validate ChemiPlex because it has been challenging to make STED multiplexably due to the special fluorophores needed to tolerate the high-energy regime and get efficiently “depleted” when illuminated with the STED beam. We made this point clearer in the main text:

“ChemiPlex resulted in a simple and effective strategy to remove the signal for all targets tested; thus, we decided to try ChemiPlex cycles under super-resolution microscopy. However, due to the need for reducing agents for signal erasure, this approach won't work under conventional dSTORM conditions, where reducing agents are needed in the buffer to help with the blinking of the fluorophores. Therefore, we turned to test ChemiPlex under the imaging challenges of Stimulated Emission Depletion (STED) super-resolution microscopy.”

More detail on the experimental pipeline is needed in each figure 1-3. Figure 1a, 2a and 3a do not provide sufficient detail of the steps involved in the process. Modifying or relocating Supp Fig 4 would be recommended.

We have added more details on the Methods and Figure legends; we have comprehensive tables for every figure, showcasing every antibody used, their dilutions, and buffers used. We think the general scheme in Fig. 1 is sufficient for a scientist to follow the logic used in every modality. Exceptionally, due to the few extra steps needed, we added a specific diagram explaining the steps used when applying ChemiPlex under STED microscopy, which is now in Supp Fig. 9.

Similarly, since ChemiPlex followed by STED imaging compromises the most significant part of the manuscript, a more detail explanation or illustrative figure would be needed to understand the full extent of the timeline and critical steps involved. The validation of the signal 'cleavage' is found in Supp Fig 8 and 9 and appears to be on a single set of images only. Some statistical justification of its efficiency is needed to be shown in the main text.

We now include more statistical analysis of the cleaving efficiency in Fig 3. Showing examples of images after erasing the signal acquired with equal parameters and displayed with the same grey-level scaling for direct visual comparison.

More generally, there is a heavy reliance upon important data being only available in the supplementary figures. Key methodological information is missing from Figures and Figure legends alike.

In agreement with the other Reviewers, we believe that our methodology is clearly explained and highly detailed. Nevertheless, we have added more detail and improved the wording for clarity in the main text, Methods, and Figure legends. We brought some of the analysis from the Supplementary to the main Figures, and we performed new statistical analysis for all modalities.

Methods lack detail: Many chemicals formula do not have subscript numbering formatted. Nearest neighbour analysis procedure is missing. Melting point procedure is lacking detail on what experiment it refers to. Buffer contents should be within methods text rather than just in Supplementary Table 2.

We have worked on all issues raised by the Reviewer (chemical formulas, NeNa, Melting point clarification, etc). Regarding the buffers suggestion: We have some of our buffers written in the main text because they appear once; however, for buffers/steps and other reagents that would need to be written repeatedly, we strongly believe that it is more efficient to link them to an organized table, stating clearly what buffer was used on which application/Figure.

Figure 1h, 2e and 3d: pseudocolouring for the individual panels is very difficult to see. How the images were acquired or processed is absent from each of these figure legends, Are these z-projections?

Finding an appropriate pallet of colors to display more than the conventional 3 colors is not trivial, but we now tried to enhance visibility. We made sure that all Figure legends clearly state the imaging technique used, and that Confocal microscopy settings are on methods with extended details. We believe, and also the journal policy demands, that the Figure legends are not a method section. Clearly, the reader should have sufficient information to understand the Figure's message, which we believe now is the case for all Figures. However, if more technical details are needed, the reader should reach the methods section, where we provide ample amount on details for an expert on the art to reproduce our results.

Figure 1c, 2c and 3c should show the comparison of the GFP Normalised intensity profile (relocation from relevant Supp Figs).

We do not understand the benefit or relevant information of doing what the Reviewer proposes. We are assessing the signal removal from the staining, which is fully independent of the "GFP" signal. We only have GFP as a reference to show that we are imaging the same region. We also clearly show the levels on the LUT, which correspond to both images displayed, making the

results directly comparable and as transparent as possible. If the reviewers refer to the “bleaching” effect, we have these controls in Supp. Fig. 5, 6 & 8 for each methodology.

Line 47-49: The logic of this sentence is unclear.

Thanks for pointing this out. We have reworded this sentence.

Line 123: U2OS-Nup96-GFP cells are not outlined in any detail. Why were they chosen?

This cell line has been highly used in the microcopy field since 2019; the benefit is clear: it has endogenously tagged nuclear pore complex with GFP and today is a standard in the microscopy field. We provide the reference where this cell line was originally described.

Line 357 (and 366): EndNote formatting error.

Thanks, we corrected this.

Line 422-424: The following statement in the discussion has not been well justified or explained:

“EnzyPlex was designed for a more straightforward implementation since 2.Nbs can be expressed as fusion proteins, including the bdSEN1 protease substrate (bdSUMO), followed by an ALFA-tag.”

Thanks for the suggestion. We have re-worded the sentence, hoping the point is now clearer:

“EnzyPlex implementation does not require chemical conjugation like in the case of OptoPlex and ChemiPlex since the ALFA tag and protease substrate (bdSUMO) can be encoded genetically and produced as fusion protein when producing the 2.Nbs”

Line 595: Italicize species name.

Yes, we have this now corrected.

Line 604: Typos. 18mm coverslips, not mM. Poly-L-Lysin’e’.

Thanks, we have corrected this typo.

Line 654: Typo. Laemmli.

Thanks, we have corrected this typo.

Line443-449: These sentences need to be re-written as the message is unclear.

We thank the Reviewer, we have decided to remove this phrase since it was indeed not contributing new insights or clarity

Line 753: Include image illumination settings.

We have now added the corresponding illumination settings.

Line 764-774: It is said that that pixel size is ‘set’ to a specific size. The technical detail on how the Confocal and STED imaging was performed is lacking.

We have clarified the imaging acquisition details further. However, we believe the level of detail provided would allow any Confocal or STED user with minimal experience to reproduce our measurements.

Reviewer #2 (Remarks to the Author):

This article presents a set of 3 different strategies to obtain multiplex imaging using sequential immunostaining. All 3 methods are based on secondary nanobodies that are cleavable in different ways, and can therefore be removed from the target (without damaging the sample) before doing another round of immunostaining. One method is optical, using UV light with an efficiency of ~70%, another one is enzymatic, with an efficiency of 90% and the last one is chemical, using TCEP with again an efficiency of 90%. All three methods are demonstrated using different imaging modalities (confocal and and super-resolution STED and dSTORM) for a variety of different targets, and for different cell lines (U2OS and primary neurons). Moreover, quantitative analysis of the spatial correlation between different synaptic markers is demonstrated. This is an important development, as the increase in multiplexed imaging requires efficient staining mechanism. Compared to DNA-based approaches, which are very powerful for single molecule imaging, these nanoplex also allows epifluorescence and confocal imaging. The article is well written, the methods section is comprehensive, and a lot of technical information is provided in the supplementary material. I think this article should be published, but I do have a few questions:

We thank the Reviewer for the positive assessment of our work.

1. Optoplex shows reduced removal efficiency compared to the other approaches, but is the easiest method to implement. I know time is typically not of the essence in these experiments, but have the authors tried using higher UV intensity? This would be particularly relevant for FRAP-like experiments on live samples, where waiting several minutes would be detrimental. Please also normalized the power of the 365nm light used by the surface illuminated to get the intensity in W/cm².

In theory, yes, and we see the point in case FRAP-like experiments are devised. However, our experience in using high-intensity irradiation is that it damages the sample, and worst for the multiplexing aim, we suspect that photochemistry is taking place, making the removal of signal much less efficient. The main scope of our work was to find a mild treatment for signal removal. It took us some effort to find the optimal spot to maximize cleavage and removal while minimizing the photo-crosslinking of the fluorophore to the sample, allowing multiplexing. Live samples will need even more optimization since living cells will also not appreciate high UV light. We have also now updated the unit of light intensity to ~60mW/cm².

2. dSTORM imaging is demonstrated using Enzyplex, but using self-blinking dyes. Can the authors comment on the compatibility with common STORM buffers (GLOXCAT or PCAPCD+Thiol).

Yes, we tried using conventional STORM buffers and dyes; however, it resulted in a much nicer reconstruction using the self-blinking dye. However, we know it works using PCA/PCD buffer + reducing agents. We chose SENP1 protease because this enzyme prefers to work in a reducing environment, and we aim for this technique to be compatible with dSTORM. We have commented on this point in the Main text and Discussion.

3. Same question but about Chemiplex: are the thiols typically used for dSTORM compatible with Chemiplex?

No, ChemiPlex will not be compatible with reducing buffers needed in dSTORM. If using self-blinking dyes, dSTORM might work under ChemiPlex regimes; however, we have not tested this directly. We have now made this point clearer in the text:

„ChemiPlex resulted in a simple and effective strategy to remove the signal for all targets tested; thus, we decided to try ChemiPlex cycles under super-resolution microscopy. However, due to the need for reducing agents for signal erasure, this approach won't work under conventional dSTORM conditions, where reducing agents are needed in the buffer to help with the blinking of the fluorophores. Therefore, we turned to test ChemiPlex under the Imaging challenges of Stimulated Emission Depletion (STED) super-resolution microscopy.“

4. In ChemiPlex, the authors modify the protocol between fig 4 and 5-6 and add unconjugated secondary nanobodies to reduce background signal, can they provide a quantification of how the background signal increases as a function of the number of cycles with and without this step?

The addition of unlabeled 2.Nbs does not reduce the background signal; it simply avoids cross-reactivity in a multiplex experiment where the same species of 1.Abs are used (e.g., from mice). An excess of unlabeled/dark 2.Nbs will simply occupy any epitopes on the 1.Abs that are free. If we do not use this blocking step, all works, but there is a risk of 2.Nbs intended to reveal target “2” that “leave” and “sit” on 1.Abs, revealing target “1” (cross signal). Therefore, unlabeled nanobodies are merely a precaution we made when using long imaging sessions (higher chances of nanobodies jumping off their bound target increases) or high-intensity illumination (STED), which might induce detaching of secondary nanobodies (and primary antibodies) by photo-modifying their epitopes.

We now provide a quantification of the remaining signals in Main and Supp. Figures for all modalities. Interestingly, the remaining signals do not increase with every new cycle.

Other comments:

- 4 color imaging is something I would consider standard, but increasing this to 5 colors (by using redder dyes, like cy7 for example) is straightforward. It would also be useful to mention in the introduction spectral-unmixing based approaches, which allow imaging of 6+fluorescent markers. [for example, Valm et al, “Applying systems level spectral imaging and analysis reveal the organelle interactome“Nature, 2017].

Yes, we know that NIR dyes like Cy7 are a straightforward option. However, we believe that set-ups and detectors for NIR are not widespread in normal laboratories. We commented on this and spectra unmixing in the introduction and discussion and added the proposed and the following citation to the Discussion (*Liu, P., Mu, X., Zhang, X.-D. & Ming, D. The Near-Infrared-II Fluorophores and Advanced Microscopy Technologies Development and Application in Bioimaging. Bioconjugate Chem. 31, 260–275 (2020)*) and *McRae, T. D., Oleksyn, D., Miller, J. & Gao, Y.-R. Robust blind spectral unmixing for fluorescence microscopy using unsupervised learning. PLoS ONE 14, e0225410 (2019).*

-Please add a reference to the newly published multiplexed DNA FASH-PAINT (Schueder et al. Cell 2024)

Yes, this is correct. We have commented on and added the reference to this work.

Typo: fix reference to {Mivolanovic2017} at the end of the results section.

Thanks, we have now corrected this.

Reviewer #3 (Remarks to the Author):

In this manuscript, Mougios et al. present a new approach for iterative immunofluorescence (IF), NanoPlex, which uses conventional primary antibodies combined with secondary nanobodies engineered for three distinct removal strategies: OptoPlex, which uses a photocleavable moiety; EnzyPlex, which uses an enzymatic cleavage; and ChemiPlex, which exploits redox chemistry to induce cleavage. While various methods have previously been developed (and are cited by the authors), NanoPlex avoids harsh chemical conditions that could disrupt sample integrity, is compatible with a range of imaging modalities (including super-resolution), uses off-the-shelf primary antibodies, and offers flexible signal removal strategies that may enable multiplexed IF in specific contexts not currently served by available alternative strategies. This is an innovative approach that is a welcome addition to current multiplexing strategies. Given that the main weakness of this approach (as noted in the manuscript by the authors) is the extent of signal removal after cleavage, more extensive quantification of this residual signal is required. The authors should address the following points related to residual signal prior to publication:

We thank the reviewer for this positive assessment of our methodology.

The authors perform similar experiments to quantify the extent of signal removal over time for each of their removal strategies in Figs 1g, 2d, and 3c. However, each of these experiments uses a different primary antibody for this experiment. It would be beneficial to the readers assessing these different approaches to see how the strategies compare for the same primary antibody.

We used different antibodies and conditions to showcase the method's flexibility. However, we can also appreciate the Reviewer's suggestion, and we have repeated the experiments using the same cells and antibodies to unify the 3 approaches and make them more comparable.

The text reports that vimentin is used to assess EnzyPlex (pg 6, line 12); however, Fig 2c-d reports tubulin staining.

Thanks for noticing this. We have changed this data set, and now the labeling is correct.

While representative images provide some indication of the extent of signal removal of each method across multiple antibodies, I would like to see a quantification of the residual signal after cleavage for all multi-round experiments similar to what was performed in Fig 4C. I am specifically referring to the experiments presented in Fig 1h, Fig 2e, Fig 2f, and Fig 3d.

We have quantified the signal removal from independent experiments and made a new analysis for each cleavage modality, now displayed in the main Figures and Supp. Figures. Additionally, we added a line profile on the exemplary set for every target to showcase the cleaving on each particular target and have a direct visual idea of the remaining signal for every target during a multiplex experiment. We hope this will help readers to understand the limitations, advantages and precautions necessary to take for applying the different signal removal modalities.

Could the authors comment on the accumulation of residual signal over successive imaging cycles? For example, in Supp Fig 9, the authors quantify residual signal in 8-ChemiPlex STED microscope after each cycle. Each cycle uses the same fluorophore, therefore residual signal from each round should remain in all subsequent cycles, if we assume that each cycle of TCEP reduction is performed to completion for each antibody (i.e., maximum signal removed). If the

authors quantified the intensity of residual signal after each cycle using the same imaging parameters, they could report on the extent of signal accumulation over successive rounds. This experiment would not necessarily have to be performed using STED.

We added now a signal removal analysis for each multiplexed image including line profiles of the exemplary images as mentioned above. As the Reviewer suggests, in the STED Fig.4c, we assess this point, and we observe that different “targets” leave a different amount of residual signal, and the cycle number seems not to be the critical aspect. When performing STED, we also performed confocal imaging, and when analyzing the residual signals (Supp. Fig. 9), we see, for example, that after cycle 6 (vimentin), the residual signal is neglectable (~0.04%). While peroxisomes had the tendency to leave more residual signals. In all cleaving modalities, the signal from peroxisomes was more resilient than the signal from filaments like Tubulin or vimentin, we can only explain this due to the membrane proximity. It is not clear if the primary antibody binds to a luminal domain on the peroxisome target proteins, making the “scape” of the cleaved fluorescent moiety, less straightforward. We have provided more quantification on residual signals for all modalities and clarified this point in the Discussion.

What is the contribution of residual signal to the colocalization analysis performed in Fig 6b-d? When comparing two proteins imaged using the same fluorophore (e.g., vGAT and gephyrin), residual signal from the first antibody should be present when imaging the second antibody and thus artifactually contribute to the quantification of colocalization. In this case (and in all other multi-cycle experiments performed in this manuscript), the authors could image the sample after cleavage and use this image to correct for residual (non-specific) signal from the subsequent cycle (after image registration to align the images). To calculate the contribution of residual signal to colocalization calculations, the authors should compare the Pearson’s r in the uncorrected images presented in the manuscript for given antibody pairs versus images in which residual signal has been subtracted, as described above.

We are aware of this potential issue with the remaining signal in our case or autofluorescence in more conventional setups. We agree with the Reviewer that if the aim would be to determine as precisely as possible the correlation between different targets, a thorough analysis of the signal remaining of each target should be performed individually for each individual target. However, to simply showcase the potential of NanoPlex, we decided to simply showcase the precision of the method by trying to “standardize” the Pearson’s correlation coefficient (PCC) in Fig. 6b. For this, we used proteins expected to correlate or to show a lack of correlation. For example, Gephyrin and PSD95 are expected not to co-localize, and to make it challenging, we imaged them with the same fluorophore in different cycles. As expected, we obtained a low PCC (~0.15) even in dense cultures of neurons where it is expected that these 2 proteins can be “close” by, especially if imaged by a diffraction-limited confocal system. If we consider that 100% of the PCC of 0.15 is due to a remaining signal from the previous cycle (which clearly is the worst-case scenario), we could estimate that ~15% of the correlation might be an “artifact” from the signal on the previous cycle. Before correcting and subtracting this exaggerated “background” correlation from all PCC analyses, we think it is more transparent to display the raw correlations obtained, having in mind the “best” and “worst” correlation obtained from proteins expected to correlate or not. We have now added this argumentation to the main text.

However, the Reviewer’s comments inspired us to perform an extra control that is now in Supp. Fig 11 (below Fig.1), where we determined the PCC from these 4 targets (vGlut, PSD95, Gephyrin, and vGAT) using confocal with conventional 2-color stainings. We did not use only

one pair of fluorophores, Atto488 and Atto643, but also Alexa568 and Atto643. We obtained equivalent PCC values as we obtained after ChemiPlex, confirming that, for example, the PCC of 0.15 between gephyrin and PSD95 (same channel in ChemiPlex) is not necessarily due to the residual signal but the complexity of the primary neurons sample and maybe the limited resolution in confocal microscopy.

We have clarify this further and included the new control which strongly suggest that ChemiPlex generates PCCs comparable to conventional imaging 2 target in 2 colors.

Fig.1 (new Supplementary Figure 11) Control of Pearson's correlation on 2 color staining neurons. Comparable staining and confocal images were made using the same antibodies as for Fig.6b, this time without cleaving fluorophores but directly comparing correlations on 2 fluorophores. vGlut was always revealed with the Nb directly labeled with Atto488 and Nanobody anti-PSD-95 with Atto643. vGAT was always revealed with 1.Ab and 2.Nb-Atto643. Geph-A is revealed with 1.Ab and 2.Nb-Atto488, while Geph-B is revealed with 1.Ab and 2.Nb-Az568. The graph represents the mean \pm SD of 6-9 images obtained from 3 independent coverslips.

Reviewer #4 (Remarks to the Author):

Thanks for the transparency, and comments combined with Reviwer's #1 suggestions.

Reviewers' Comments:

Reviewer #1:

Remarks to the Author:

The authors have addressed all my concerns and queries. The updated version is in my view excellent and ready for publication.

Reviewer #2:

Remarks to the Author:

I think the authors did a good job addressing my (and the other reviewers') questions, and made it clearer in this revised manuscript what the different advantages and drawbacks of the 3 methods were. I still think that a supplementary figure using "normal dSTORM" and Enzyplex would have been nice since it seems the data is already there, but this is not worth another round of revisions.

I did however notice a few typos/unclear sentences in the revised parts of the manuscript that would need fixing:

- l350 : missing "As" at the beginning of the sentence ?
- l479-480 "being optoplex the least efficient"
- l482 : les optimal
- l486 : microtubuli
- l801: missing ")"

Reviewer #3:

Remarks to the Author:

The authors have addressed my concerns commendably and I recommend this work for publication.

Reviewer #4:

Remarks to the Author:

The authors have addressed all of our comments and the updated version of the manuscript is of high quality.

Reviewer #1 (Remarks to the Author):

The authors have addressed all my concerns and queries. The updated version is in my view excellent and ready for publication.

We thank the reviewer for the positive assessment.

Reviewer #2 (Remarks to the Author):

I think the authors did a good job addressing my (and the other reviewers') questions, and made it clearer in this revised manuscript what the different advantages and drawbacks of the 3 methods were. I still think that a supplementary figure using "normal dSTORM" and Enzyplex would have been nice since it seems the data is already there, but this is not worth another round of revisions.

We thank the reviewer for the positive assessment.

I did however notice a few typos/unclear sentences in the revised parts of the manuscript that would need fixing:

- l350 : missing "As" at the beginning of the sentence ?
- l479-480 "being optoplex the least efficient"
- l482 : les optimal
- l486 : microtubuli
- l801: missing ")"

We thank the reviewer for finding these issues. We have now addressed them all.

Reviewer #3 (Remarks to the Author):

The authors have addressed my concerns commendably and I recommend this work for publication.

We thank the reviewer for the positive assessment.

Reviewer #4 (Remarks to the Author):

The authors have addressed all of our comments and the updated version of the manuscript is of high quality.

We thank the reviewer for the positive assessment.